# Landfill leachate: An invisible threat to soil quality of temperate Himalayas

Shayesta Islam[1☯], Haleema Bano[1☯], Asif Aziz Malik[2]*, Fahad Alotaibi[3]

**1** Division of Environmental Sciences, SKUAST K, Srinagar, Jammu and Kashmir, India, **2** Division of Basic Science and Humanities, SKUAST K, Srinagar, Jammu and Kashmir, India, **3** Department of Soil Science, College of Food and Agriculture Sciences, King Saud University, Riyadh, Saudi Arabia

☯ These authors contributed equally to this work.
* drasifskuast@gmail.com

**Data Availability Statement:** All relevant data are within the manuscript and its Supporting Information files.

**Funding:** Funder: Fahad Naseer Alotaibi, Assistant professor- Department of Soil Science, College of

## Abstract

Landfills are the most affordable and popular method for managing waste in many parts of the world, However, in most developing nations, including India, the infiltration of hazardous materials from improperly managed dumping site continues to be a significant environmental problem. Around the world, leachate is a significant point source of contamination in numerous environmental media, including soil, groundwater, and surface water. Soil is an important asset as it is the key factor for food production and has tremendous significance in achieving sustainable development goals (SDGs). The contaminants from soil enter into food chain and ultimately reach humans. So in order to prevent the adverse effects of toxic elements on humans, there is need to maintain the soil quality and to prevent deterioration. Keeping in view the consequences of unscientific management of waste, the goal of the experiment was to determine how landfill leachate from Achan landfill affected the soil quality in the temperate Himalayas. All four seasons of the year, viz Spring, Summer, Autumn, and Winter, at four sites viz, Center of dumping site, inside, Outside and Control were monitored. Among sites center was found to have maximum value of EC (3.04 dS/m), Moisture content (42.51%), N (285.43 mg/kg), P (70.07 mg/kg), K (265.71 mg/kg), Ca (957.67 mg/kg), Mg(402.42 mg/kg), Zn (2.02 mg/kg), Fe (10.56 mg/kg), Cu (2.07 mg/kg), Mn (10.73 mg/kg), Pb (85.02 mg/kg), Cd (4.50 mg/kg), Ni (29.04 mg/kg), Cr (23.37 mg/kg), As (14.10 mg/kg). While as the lowest value of all parameters was reported at control site. From the study it is recommended that the waste generated is mostly organic (65–75%), thus can be segregated and treated at source. The waste can be treated at source using microbial consortium technology in order to transform the waste in to wealth in a sustainable way and to prevent the deterioration of soil quality.

## Introduction

Soil ecosystems are the foundation for food production and are crucial for achieving sustainable development goals (SDGs), so are regarded as significant environmental assets [1]. It supports or carries out a variety of tasks, such as biomass production, storing, filtering, and

Food and Agriculture Sciences, King Saud University, P.O.Box 2460, Riyadh, Saudi Arabia. Project Number: RSPD2024R889 "In addition to funding the funder has a role in preparation and modification of manuscript".

**Competing interests:** The authors have declared that no competing interests exist.

changing a wide variety of materials, including water and nutrients [2]. The majority of the carbon (C) and nutrient elements that support life are generated by soils, which also act as a repository for them [3]. Soils are also the pedestal that plants rely on to remain upright. They also serve as the habitat for a vast biodiversity and biomass of soil organisms [4]. Soils hold the water that plants and soil creatures need to exist and thrive, and they restrict the pace of water movement, limiting erosion and soil loss [5]. It has been reported that improving soil health through various amendments like biochar application improves soil physicochemical properties and enzymatic activities [6]. But this valuable asset is currently under serious threat due to generation of excessive quantity of waste due to rapid population growth and urbanization. According to the reports of world bank, the world wide generation of municipal solid waste in 2018 was 1.3 billion tons per year and is expected to reach approximately 2.2 billion tons by 2025 [7]. Kashmir Valley is also grappling with significant challenges stemming from increasing waste generation. Srinagar city alone produces 0.526 kg of waste per capita per day, with Anantnag following at 0.479 kg, Ganderbal at 0.400 kg, and Budgam at 0.397 kg respectively. The total annual waste generated across these districts amounts to 57,199.99 Metric Tonnes (MT), with Srinagar producing the highest at 236,732.75 MT and Budgam the lowest at 42,840.00 MT [8]. This enormous amount of waste can pose great challenge especially to low- and middle-income countries and developing nations.

The immense volume of waste generated from various sources ultimately finds its way to landfills, with Srinagar city in the Kashmir Valley being no exception. Here, all the waste produced across the city is disposed of at the Achan landfill site, the sole dumping ground for Srinagar. This site, located in the northern part of the city, between the coordinates 34° 09′ N "latitude" and 74° 79′ E "longitude", lies approximately 5–6 kilometers from the city's center. Before 1985, Srinagar produced minimal waste, free from the burden of plastic. Back then, the Srinagar Municipal Corporation (SMC) disposed of the city's waste either in the Noorbagh grounds or along the banks of the river Jhelum. However, with the advent of stricter environmental policies, the Government of Jammu and Kashmir, following state cabinet approval, transferred around 30.35 hectares (75 acres) of state land to the SMC for waste disposal at Achan [9]. By 1986, this site became the primary location for the open dumping of municipal solid waste. For many years, vast amounts of refuse were deposited here without consideration for the environmental consequences. Achan remained the sole dumping ground for Srinagar until 2008 [10], with waste from 518 collection points across the city being transported to this location [9]. Once a thriving wetland, home to vibrant birds and lush vegetation, Achan has since become a source of pollution, contaminating water bodies, air, and soil. The nearby population has also been affected, suffering from various diseases. The waste deposited at the site undergoes biodegradation through a combination of physical, chemical, and biological processes, leading to the production of leachate—a dark brown liquid with a foul odor. This leachate contains dissolved organic and inorganic compounds, nutrients, suspended particles, heavy metals, and hazardous chemicals. When left untreated and uncontrolled, it poses a serious threat to natural and agricultural ecosystems, significantly degrading soil quality due to its high concentrations of nutrients, heavy metals, and soluble salts [11,12].

Contaminated soil affects ecosystems and human, plant and animal health in a number of ways [13]. The acidic leachate formed due to production of organic acids liberated by the degradation of waste, leads to the acidic surroundings that destroys micro-organisms, which are helpful in improving the composition of the soil. Furthermore contaminated soil has the potential to alter agricultural metabolism and diminish crop yields by forcing trees and crops to absorb soil pollutants [14]. In addition to this, contaminated soils with high nitrogen and phosphorus levels will leach into waterways, leading in death of aquatic organisms by diminishing the amount of dissolved oxygen [15]. The metals in leachate are common

environmental pollutants that are non-biodegradable, can deplete available soil resources, and have a negative impact on plant growth and yield [16,17].

Keeping in view the far reaching consequences of landfill leachate and contaminated soil and need for global attention on improving or restoring soil health, it is imperative to evaluate the affect of leachate on soil physico chemical properties for proper management strategy to prevent soil degradation and alleviate the threats posed by the expansion of waste. Understanding and estimating the quality of leachate-impacted soils can establish an opportunity to judge the sustainability of land management and land-use systems. In this context, the present study aimed to evaluate the influence of landfill leachate from Srinagar's Achan dump on soil physico-chemical parameters across different seasons and sites.

## Study area

The study was conducted at the Achan landfill, located in Srinagar city, Jammu and Kashmir UT, at an elevation of 1600 meters above mean sea level. The landfill's coordinates range between 74˚41' 6" and 74˚57' 27" East Longitude and 33˚ 59' 14" and 34˚12' 37" North Latitude, as shown in Fig 1.

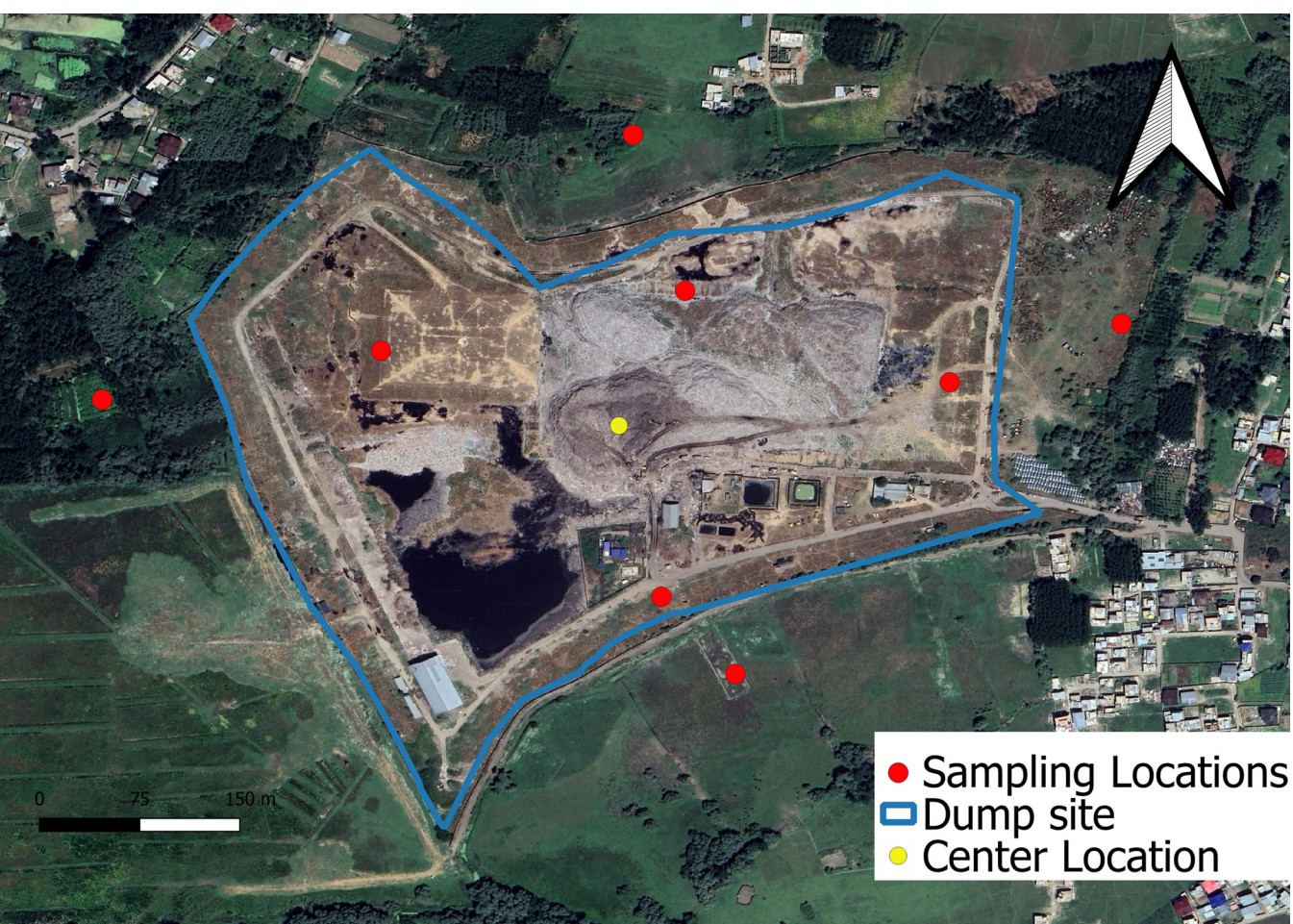

**Fig 1. Map of study area (Map prepared by USGS).**

Spanning an area of 30.35 hectares (75 acres), the landfill is situated amidst residential settlements and water bodies, directly impacting the lives of residents in nine villages: Saidapora, Shonglipora, Waganpora, Sangam, Braywar, Danmar, Guzerbal, Noorshah Colony, and Bagh-i-Lal Pandith. Every day, the Srinagar Municipal Corporation (SMC) disposes of 450 metric tonnes in this site. Waste generation is projected to surge to 1,723 metric tonnes per day by 2035. Currently, 62% of the total waste produced daily in Srinagar consists of organic material, with 7% being plastic waste [18]. Approximately 65–70% of Srinagar's municipal solid waste is collected through door-to-door methods and street bin systems, and then transported to the Achan landfill. However, the remaining 30–35% is illegally discarded into depressions, river embankments, open spaces, or is burned locally by individuals and Safai Karamcharis. This not only causes a public nuisance but also creates breeding grounds for various diseases [19]. Due to inadequate waste management and improper handling, large heaps of waste have accumulated at the site, emitting a foul odor that disturbs nearby residents. During the rainy season, leachate from the waste seeps into the soil, altering its physico-chemical properties and rendering the surrounding land unsuitable for agriculture.

## Materials and methods

Soil samples were collected from the Achan dumpsite in Srinagar throughout all four seasons of 2022—spring, summer, autumn, and winter—following proper authorization from the Srinagar Municipal Corporation, headquartered at Kara Nagar, Srinagar, Jammu and Kashmir. Four distinct sampling sites were chosen: the center of the dumpsite, inside the dumpsite (four cardinal points within), outside the dumpsite (four cardinal points outside), and a control site located at the Shalimar campus of SKUAST Kashmir. These sites were selected to assess the impact of leachate produced from the waste on soil properties at varying distances from the dumping points. Three replicates were collected from each site.

Samples were taken solely from the soil, as leachate was directly seeping into the ground. The soil was collected in separate moisture cans to preserve its moisture content. Once collected, the samples were air-dried in the shade, then ground and sieved through a 2mm mesh before analyzing various physico-chemical parameters. Soil moisture content was determined using the gravimetric method by Prihar and Sandhu [20], while soil pH and electrical conductivity were measured using the potentiometric method with a pH meter and an electrical conductivity meter [21]. Nitrogen content was determined via the potassium permanganate method [22], and the available phosphorus was measured using the spectrophotometric method by Olsen et al. [23]. The Jackson method [21] was used to quantify available potassium, while calcium and magnesium levels were assessed using the Versenate method, as outlined by Chesin and Yein [24]. Heavy metal concentrations were determined using the DTPA extraction process and an atomic absorption spectrophotometer (AAS), following the method of Lindsay and Norwell [25].

## Statistical analysis

The data in the table are given as an interactive mean of sites and seasons in accordance with the usual technique outlined by Gomez and Gomez [26]. Two-way ANOVA (analysis of variance) was performed at a 5% level of significance to test for differences across sites and seasons. In the univariate statistical pairwise comparison between all the samples comparing the means of two populations that are seasons and sites in our situations, analysis of variance with post hoc testing (Duncan's multiple range test) is performed [27]. The significance is shown by the various letters, and vice versa [27].

## Results

pH determines the acidity or alkalinity of soil solutions and refers to the soil's hydrogen ion content. In our study among seasons maximum mean pH was recorded in winter (6.6), followed by spring (6.5), autumn (6.4) and minimum was recorded in summer season (6.3). Among sites maximum mean pH was recorded in control (6.7), outside (6.6), inside (6.3) and minimum (6.2) was recorded at center as depicted in Table 1. The pH values varied significantly between summer, autumn, and winter, but there was no significant difference between spring and autumn. Additionally, the pH values at different locations—inside, outside, and control—were significantly different, though there was no significant difference between the center and inside locations. During this study mean EC of dumpsite soil was found to deviate from 0.27 dS/m to 3.43 dS/m. Maximum mean value of EC among seasons was recorded in summer (1.81 dS/m), spring (1.66 dS/m), autumn (1.48 dS/m) and minimum was recorded in winter (1.35 dS/m). Among sites maximum mean value of EC was recorded at center (3.04 dS/m), followed by inside (2.01 dS/m), outside (0.86 dS/m) and minimum was recorded at control (0.40 dS/m). The EC values during spring, summer, autumn and winter were significantly different. In addition to this EC values at all sites were significantly different. In this study, the mean moisture content values ranged from 12.87% to 48.42%. Moisture content was recorded maximum in winter (36.95%) among seasons, followed by spring (32.48%), autumn (29.34%) and minimum was recorded in summer (25.23%). Among sites mean maximum moisture content was recorded at center (42.51%), followed by inside (34.07%), outside (28.83%) and minimum was recorded at control site (18.58%). The moisture content values between all seasons were significantly different, in addition this, the moisture content values between all sites were also significantly different.

**Table 1. Seasonal and site wise variation in pH, EC (dS/m) and moisture content (%) of soil.**

| Parameters | Seasons | Sites | | | | Mean | C.D (P≤0.05) |
|---|---|---|---|---|---|---|---|
| | | Center | Inside | Outside | (Control) | | |
| **pH** | Spring | $6.3\pm0.05^{fgh}$ | $6.4\pm0.09^{efg}$ | $6.6\pm0.06^{bc}$ | $6.8\pm0.10^{ab}$ | **6.5<sup>b</sup>** | Sites: 0.12 Seasons: 0.12 Sites × Seasons: 0.24 |
| | Summer | $6.1\pm0.05^{h}$ | $6.2\pm0.11^{gh}$ | $6.4\pm0.06^{cdef}$ | $6.5\pm0.14^{bcde}$ | **6.3<sup>c</sup>** | |
| | Autumn | $6.2\pm0.35^{gh}$ | $6.3\pm0.09^{fgh}$ | $6.5\pm0.04^{bcd}$ | $6.7\pm0.18^{ab}$ | **6.4<sup>b</sup>** | |
| | Winter | $6.4\pm0.21^{defg}$ | $6.5\pm0.13^{cdef}$ | $6.7\pm0.05^{ab}$ | $6.9\pm0.15^{a}$ | **6.6<sup>a</sup>** | |
| | **Mean** | **6.2<sup>c</sup>** | **6.3<sup>c</sup>** | **6.6<sup>b</sup>** | **6.7<sup>a</sup>** | | |
| **EC** | Spring | $3.16\pm0.28^{ab}$ | $2.12\pm0.23^{cd}$ | $0.95\pm0.24^{ef}$ | $0.41\pm0.10^{gh}$ | **1.66<sup>ab</sup>** | Sites: 0.21 Seasons: 0.21 Sites× Seasons:0.42 |
| | Summer | $3.43\pm0.25^{a}$ | $2.23\pm0.29^{c}$ | $1.06\pm0.24^{e}$ | $0.55\pm0.22^{fgh}$ | **1.818<sup>a</sup>** | |
| | Autumn | $2.84\pm0.21^{b}$ | $1.96\pm0.19^{cd}$ | $0.79\pm0.20^{efg}$ | $0.35\pm0.13^{h}$ | **1.485<sup>bc</sup>** | |
| | Winter | $2.74\pm0.26^{b}$ | $1.74\pm0.51^{d}$ | $0.66\pm0.25^{efgh}$ | $0.27\pm0.06^{h}$ | **1.35<sup>c</sup>** | |
| | **Mean** | **3.04<sup>a</sup>** | **2.01<sup>b</sup>** | **0.86<sup>c</sup>** | **0.40<sup>d</sup>** | | |
| **Moisture content (%)** | Spring | $43.34\pm1.01^{b}$ | $36.48\pm0.87^{d}$ | $29.44\pm0.47^{g}$ | $20.67\pm1.01^{j}$ | **32.48<sup>b</sup>** | Sites:1.797 Seasons: 1.797 Sites× Seasons:3.594 |
| | Summer | $36.87\pm0.99^{d}$ | $26.65\pm0.99^{h}$ | $24.55\pm0.97^{i}$ | $12.87\pm0.69^{l}$ | **25.23<sup>d</sup>** | |
| | Autumn | $41.43\pm1.01^{c}$ | $31.37\pm1.03^{f}$ | $28.09\pm0.99^{gh}$ | $16.48\pm1.01^{k}$ | **29.34<sup>c</sup>** | |
| | Winter | $48.42\pm1.01^{a}$ | $41.79\pm1.02^{bc}$ | $33.27\pm1.00^{e}$ | $24.32\pm1.01^{i}$ | **36.95<sup>a</sup>** | |
| | **Mean** | **42.51<sup>a</sup>** | **34.07<sup>b</sup>** | **28.83<sup>c</sup>** | **18.58<sup>d</sup>** | | |
| Standard limits (ICAR manual) | | pH Acidic < 6.0 Normal to saline (6.0–8.5) Tending to became alkaline (8.9–9.0) Alkaline > 9.0 | | EC Low <1.0 dS/m Medium (1.0–3.0 dS/m) High > 3.0 dS/m | | | Moisture content Low (3–10%) Medium (20–40%) High > 40% | |

Mean N concentration in soils of the experimental site ranged from 110.67 to 320.49 (mg/kg). The concentration of N was maximum at center of dumping site (285.43 mg/kg), followed by inside (266.73 mg/kg), outside (241.61 mg/kg) and least (131.26 mg/kg) was reported at control site during all the seasons. Among seasons the maximum N concentration was observed during summer season (264.11 mg/kg), followed by spring season (244.93 mg/kg), autumn season (221.85 mg/kg) and least (194.14 mg/kg) was observed in winter season as depicted in Table 2. Additionally, the mean values of N between sites change significantly, and the mean values of N between all seasons were significantly different.

Soil phosphorus content at the experimental site ranged from 15.38 to 87.36 mg/kg. The highest mean P level was found at the center of the dumping site (70.07 mg/kg), followed by the inside (55.12 mg/kg) and outside (41.16 mg/kg) locations, with the lowest at the control site (22.16 mg/kg), as shown in Table 2. Seasonally, the highest mean phosphorus content was observed during the summer (60.98 mg/kg), followed by spring (49.51 mg/kg), autumn (41.85 mg/kg), and the lowest during winter (36.15 mg/kg). Additionally, there are considerable differences in the mean values of P between locations and the mean values of P throughout all seasons.

Soil potassium levels ranged from 96.54 to 301.28 mg/kg. Among seasons maximum mean potassium (239.70 mg/kg) was recorded in summer season followed by spring (222.63 mg/kg), autumn (204.14 mg/kg) and minimum (183.58 mg/kg) was recorded in winter season. Among sites mean maximum value of potassium was recorded in center (265.71 mg/kg), followed by inside (242.35 mg/kg), outside (225.85 mg/kg) and minimum (116.16 mg/kg) was recorded at control site as depicted in Table 2. Additionally, there are considerable differences in the mean values of K between sites as well as the mean values of K across all seasons.

During the present study, mean Ca concentration ranged from 668.33 to 984.39 (mg/Kg). Mean maximum concentration of Calcium among seasons was recorded in summer season

**Table 2. Seasonal and site wise variation in N, P and K (mg/Kg) of soil.**

| Parameters | Seasons | Sites | | | | Mean | C.D (P≤0.05) |
|---|---|---|---|---|---|---|---|
| | | **Center** | **Inside** | **Outside** | **Control** | | |
| **N** | Spring | $300.33\pm2.43^c$ | $283.40\pm2.74^d$ | $256.73\pm2.06^e$ | $139.27\pm3.13^k$ | **$244.93^b$** | Sites: 4.700 |
| | Summer | $320.49\pm1.80^a$ | $307.15\pm0.65^b$ | $282.34\pm1.95^d$ | $146.47\pm1.16^j$ | **$264.11^a$** | Seasons: 4.700 |
| | Autumn | $280.41\pm1.79^d$ | $252.50\pm2.19^f$ | $225.85\pm2.08^h$ | $128.64\pm2.07^l$ | **$221.85^c$** | Sites × Seasons: 9.401 |
| | Winter | $240.51\pm2.04^g$ | $223.86\pm1.52^h$ | $201.54\pm1.32^i$ | $110.67\pm2.24^m$ | **$194.14^d$** | |
| | **Mean** | **$285.43^a$** | **$266.73^b$** | **$241.61^c$** | **$131.26^d$** | | |
| **P** | Spring | $71.29\pm1.12^c$ | $57.42\pm0.89^e$ | $43.95\pm1.71^h$ | $25.39\pm1.08^k$ | **$49.51^b$** | Sites: 2.017 |
| | Summer | $87.36\pm0.85^a$ | $73.44\pm1.87^b$ | $54.68\pm1.24^f$ | $28.45\pm2.01^l$ | **$60.98^a$** | Seasons: 2.017 |
| | Autumn | $63.35\pm1.04^d$ | $48.60\pm1.24^g$ | $36.06\pm1.23^j$ | $19.42\pm0.95^m$ | **$41.86^c$** | Sites × Seasons: 4.034 |
| | Winter | $58.29\pm1.02^e$ | $41.01\pm41.01^i$ | $29.94\pm0.54^k$ | $15.38\pm1.08^n$ | **$36.15^d$** | |
| | **Mean** | **$70.07^a$** | **$55.12^b$** | **$41.16^c$** | **$22.16^d$** | | |
| **K** | Spring | $275.70\pm0.38^b$ | $251.77\pm0.97^d$ | $237.70\pm1.08^e$ | $125.35\pm1.03^l$ | **$222.63^b$** | Sites: 6.887 |
| | Summer | $301.28\pm1.03^a$ | $273.33\pm1.05^c$ | $251.77\pm1.63^d$ | $132.41\pm1.12^k$ | **$239.70^a$** | Seasons: 6.887 |
| | Autumn | $253.38\pm1.12^d$ | $234.29\pm0.90^f$ | $218.64\pm1.12^h$ | $110.35\pm0.94^m$ | **$204.14^c$** | Sites × Seasons:13.774 |
| | Winter | $232.49\pm1.07^g$ | $210.03\pm1.03^i$ | $195.28\pm0.95j$ | $96.54\pm1.11^n$ | **$183.58^d$** | |
| | **Mean** | **$265.71^a$** | **$242.35^b$** | **$225.85^c$** | **$116.16^d$** | | |
| Standard Limits (ICAR manual) | | **N**<br>Low<107.1 mg/kg<br>Medium (107.1–214.2 mg/kg)<br>High >214.2 mg/kg | | **P**<br>Low < 13.2mg/kg<br>Medium (13.2–33 mg/kg)<br>High > 33 mg/kg | | **K**<br>Low < 49.10 mg/kg<br>Medium (49.10–125 mg/kg)<br>High > 125 mg/kg | |

N: Nitrogen, P: Phosphorus, K: Potassium.

**Table 3. Seasonal and site wise variation in Ca and Mg (mg/kg) of soil.**

| Parameters | Seasons | Sites | | | | Mean | C.D (P$\leq$0.05) |
|---|---|---|---|---|---|---|---|
| | | Center | Inside | Outside | Control | | |
| Ca | Spring | 982.31±0.96[c] | 965.34±0.97[d] | 893.60±0.99[i] | 723.50±0.94[n] | **893.44[b]** | Sites:1.809 |
| | Summer | 997.74±1.13[a] | 984.39±0.90[b] | 913.43±0.93[g] | 758.51±0.88[m] | **913.52[a]** | Seasons:1.809 |
| | Autumn | 940.40±1.06[e] | 925.85±0.73[f] | 860.11±0.94[l] | 701.31±1.10[°] | **856.92[c]** | Sites × Seasons:3.618 |
| | Winter | 910.25±0.96[h] | 889.40±1.08[j] | 838.37±1.04[k] | 668.33±0.95[p] | **826.58[d]** | |
| | **Mean** | **957.67[a]** | **941.24[b]** | **876.38[c]** | **712.91[d]** | | |
| Mg | Spring | 412.86±0.71[b] | 405.61±1.35[c] | 376.28±0.81[g] | 303.89±1.25[l] | **374.66[b]** | Sites:3.873 |
| | Summer | 419.24±0.97[a] | 418.61±0.93[a] | 383.85±1.02[f] | 318.66±1.21[k] | **385.09[a]** | Seasons:3.873 |
| | Autumn | 395.12±0.77[d] | 388.99±0.90[e] | 352.50±1.27[j] | 294.61±1.13[m] | **357.80[c]** | Sites × Seasons:7.745 |
| | Winter | 382.47±0.95[f] | 373.67±0.94[h] | 361.38±1.03[i] | 280.82±1.17[n] | **349.58[d]** | |
| | **Mean** | **402.42[a]** | **396.72[b]** | **368.48[c]** | **299.49[d]** | | |
| Standard limits (ICAR manual) | | Ca<br>Low< 500 mg/kg<br>Medium (500–2500 mg/kg)<br>High > 2500 mg/kg | | | Mg<br>Sandy soils (51–250 mg/kg)<br>Clayey soils (101–500 mg/kg) | | |

Ca: Calcium, Mg: Magnesium.

(913.52 mg/kg), followed by spring (893.44 mg/kg), autumn (856.92) and minimum was recorded in winter (826.58). Among sites mean maximum concentration of calcium was recorded in center (957.67 mg/kg), inside (941.24 mg/kg), outside (876.38 mg/kg) and minimum (712.91 mg/kg) was recorded at control. The mean values of Ca between all seasons were significantly different; furthermore the mean values of Ca between sites also vary significantly.

During the present study the mean Mg concentration ranged from 280.82 to 418.61 (mg/Kg). Magnesium concentration was recorded maximum (385.09 mg/kg) during summer season, followed by spring (374.66 mg/kg), autumn (357.80 mg/kg) and minimum (349.58 mg/kg) was recorded in winter. Among sites maximum mean concentration of magnesium was recorded in center (402.42 mg/kg), followed by inside (396.72 mg/kg), outside (368.48 mg/kg) and minimum (299.49 mg/kg) was recorded at control site as depicted in Table 3. Additionally, there are considerable differences in the mean levels of Mg between locations as well as the mean values of Mg across all seasons.

Zinc (Zn) concentration in the soil in the current study ranged from 0.75 to 3.57 (mg/Kg). Among seasons maximum mean concentration of Zn (2.89 mg/kg) was recorded in winter season, followed by autumn (2.54 mg/kg), spring (1.75 mg/kg) and minimum was recorded in summer (1.53 mg/kg). Among sites maximum mean concentration of Zn was recorded at center (2.92 mg/kg), followed by inside (2.50 mg/kg), outside (2.03 mg/kg) and minimum (1.27 mg/kg) was recorded at control. The mean values of Zn between spring, summer and autumn were significantly different, while as no significant difference was reported between autumn and winter. However the mean values of all sites were significantly different. The mean maximum Fe in the current study ranged from 1.51 to 12.51 (mg/Kg). For iron (Fe) maximum mean concentration among seasons was recorded in winter (8.78 mg/kg), followed by autumn (8.26 mg/kg), spring (5.48 mg/kg) and minimum (4.65 mg/kg) was recorded in summer, while as among sites maximum mean concentration was recorded at center (10.56 mg/kg), followed by inside (6.52 mg/kg), outside (6.37 mg/kg) and minimum (3.71 mg/kg) was recorded at control. The mean values of Fe between all seasons were significantly different. Among sites the values of Fe were significantly different between center, inside and control, however no

significant difference was reported between inside and outside dumpsite. Mean concentration of Cu ranged from 0.17 to 2.72 (mg/Kg). Copper (Cu) was recorded maximum among seasons in winter (2.11 mg/kg), followed by autumn (1.97 mg/kg), spring (1.19 mg/kg) and minimum was recorded in summer (0.88 mg/kg). Among sites maximum mean concentration of Cu was recorded in center (2.07 mg/kg), followed by inside (1.80 mg/kg), outside (1.74 mg/kg) and minimum was recorded at control (0.54 mg/kg). The mean values of Cu were significantly different throughout all seasons, and it was also observed that these differences were significant between sites.

Mean Manganese (Mn) concentration of soils ranged from 4.54 to 11.83 (mg/Kg) shown in Table 4. Manganese was recorded maximum in winter (8.92 mg/kg), followed by autumn (8.38 mg/kg), spring (7.62 mg/kg) and minimum was recorded in summer (7.24 mg/kg). Among sites maximum mean concentration was recorded at center (10.73 mg/kg), followed by inside (9.34 mg/kg), outside (7.42 mg/kg) and minimum was recorded at control (4.68 mg/kg) as depicted in Table 4. The mean Mn values also varied significantly between different locations and across all seasons.

In the present study the concentration of lead (Pb) ranged from 0.98 mg/kg to 98.3 mg/kg. The mean maximum concentration of lead (Pb) among seasons was recorded in summer (72.1 mg/kg), followed by autumn (68.5 mg/kg), spring (59.08 mg/kg) and minimum was recorded in winter (35.9 mg/kg) while as among sites maximum mean concentration of Pb was recorded

**Table 4. Seasonal and site wise variation in micronutrients, viz Zn, Fe, Cu and Mn (mg/kg) of soil.**

| Parameters | Seasons | Sites | | | | Mean | C.D (P≤0.05) |
|---|---|---|---|---|---|---|---|
| | | Center | Inside | Outside | Control | | |
| **Zn** | Spring | 2.39±0.14[de] | 2.01±0.31[ef] | 1.55±0.33[gh] | 1.07±0.28[ij] | **1.75[b]** | Sites:0.212 Seasons:0.212 Sites × Seasons:0.424 |
| | Summer | 2.34±0.52[de] | 1.76±0.01[fg] | 1.30±0.04[hi] | 0.75±0.32[j] | **1.53[c]** | |
| | Autumn | 3.36±0.21[ab] | 2.99±0.09[bc] | 2.53±0.17[d] | 1.27±0.12[hi] | **2.54[a]** | |
| | Winter | 3.57±0.29[a] | 3.26±0.15[ab] | 2.74±0.24[cd] | 1.98±0.24[ef] | **2.89[a]** | |
| | **Mean** | **2.92[a]** | **2.50[b]** | **2.03[c]** | **1.27[d]** | | |
| **Fe** | Spring | 9.25±0.29[b] | 5.21±0.15[f] | 5.14±0.32[f] | 2.31±0.14[h] | **5.48[c]** | Sites:0.250 Seasons:0.250 Sites × Seasons:0.50 |
| | Summer | 8.45±0.21[c] | 4.41±0.24[g] | 4.24±0.23[g] | 1.51±0.18[i] | **4.65[d]** | |
| | Autumn | 12.05±0.17[a] | 8.01±0.27[cd] | 7.85±0.11[d] | 5.13±0.19[f] | **8.26[b]** | |
| | Winter | 12.51±0.89[a] | 8.46±0.15[c] | 8.27±0.18[cd] | 5.88±0.06[e] | **8.78[a]** | |
| | **Mean** | **10.56[a]** | **6.52[b]** | **6.37[b]** | **3.71[c]** | | |
| **Cu** | Spring | 1.62±0.24[d] | 1.36±0.09[e] | 1.29±0.13[e] | 0.49±0.12[h] | **1.19[c]** | Sites:0.117 Seasons:0.117 Sites × Seasons: 0.235 |
| | Summer | 1.33±0.11[e] | 1.04±0.08[f] | 0.99±0.11[f] | 0.17±0.06[i] | **0.88[d]** | |
| | Autumn | 2.61±0.24[ab] | 2.33±0.15[c] | 2.28±0.18[c] | 0.65±0.05[gh] | **1.97[b]** | |
| | Winter | 2.72±0.06[a] | 2.46±0.07[bc] | 2.39±0.19[bc] | 0.87±0.10[fg] | **2.11[a]** | |
| | **Mean** | **2.07[a]** | **1.80[b]** | **1.74[b]** | **0.54[c]** | | |
| Mn | Spring | 10.21±0.63[cd] | 8.79±0.25[f] | 6.89±0.39[i] | 4.58±0.17[j] | **7.62[c]** | Sites: 0.212 Seasons: 0.212 Sites × Seasons: 0.425 |
| | Summer | 9.73±0.04[e] | 8.32±0.28[g] | 6.47±0.08[i] | 4.54±0.05[j] | **7.24[d]** | |
| | Autumn | 11.18±0.23[b] | 9.83±0.11d[e] | 7.81±0.39[h] | 4.69±0.06[j] | **8.38[b]** | |
| | Winter | 11.83±0.06[a] | 10.42±0.09[c] | 8.53±0.21[fg] | 4.92±0.05[j] | **8.92[a]** | |
| | **Mean** | **10.73[a]** | **9.34[b]** | **7.42[c]** | **4.68[d]** | | |
| Standard limits (ICAR manual) | | **Zn** Low < 0.5 mg/kg Medium (1–3 mg/kg) High > 5 mg/kg | | **Fe** Low < 2 mg/kg Medium (4 -6mg/kg) High > 10 mg/kg | | **Cu** Low < 0.1 mg/kg Medium (0.3–0.8mg/kg) High > 3 mg/kg | Mn Low< 0.5 mg/kg Medium (1.2–3.5 mg/kg) High > 6 mg/kg |

Zn: Zinc, Fe: Iron, Cu: Copper, Mn: Manganese.

at center (85.02 mg/kg), followed by inside (76.02 mg/kg), outside (68.72 mg/kg) and minimum (5.90 mg/kg) was recorded at control. Additionally to the fact that the mean Pb values between sites varied greatly, these mean values of Pb between seasons varied significantly as well. The concentration of cadmium Cd ranged from 0.06mg/kg to 5.89 mg/kg. Cadmium (Cd) was recorded maximum in summer season (4.46 mg/kg), followed by autumn (3.80 mg/kg), spring (3.22 mg/kg) and minimum (1.57 mg/kg) was recorded in winter and among maximum mean concentration of Cd was recorded at center (4.56 mg/kg), followed by inside (4.26 mg/kg), outside (4.08 mg/kg) and minimum was recorded at control (0.15 mg/kg). Among the sites, the mean cadmium values were significantly different between the center and control, and inside and control; however, no significant difference was reported between outside and control. The concentration of nickel (Ni) ranged from 0.65mg/kg to 42.10 mg/kg. Ni was recorded maximum in summer (29.70 mg/kg), followed by autumn (22.23 mg/kg), spring (17.01 mg/kg) and minimum was recorded in winter (9.30 mg/kg). In case of sites it was recorded maximum at center (29.04 mg/kg), followed by inside (25.86 mg/kg), outside (21.88 mg/kg) and minimum (1.47 mg/kg) was recorded at control as depicted in Table 5. The mean values of Ni between seasons as well as sites differ significantly. The concentration of chromium (Cr) ranged from 2.31 mg/kg to 30.40 mg/kg. Among seasons max concentration (22.41 mg/kg) was reported during summer season followed by autumn (19.67 mg/kg), spring (15.37 mg/kg) and min during winter (9.88 mg/kg). Among sites maximum concentration (23.37 mg/kg) was reported at center of dumpsite, followed by inside (21.12 mg/kg), outside (18.61 mg/kg) and minimum was reported at control site (4.23 mg/kg). The mean values of Cr between seasons were reported to be significantly different. Among sites the mean values of Cr between inside, outside, control were significantly different; however no significant difference in mean values of Cr was reported between center and inside. The value of Arsenic (As) ranged from 0.23 mg/kg to 18.50 mg/kg. Among seasons max concentration (12.63 mg/kg) was reported during summer season followed by autumn (9.86 mg/kg), spring (8.04 mg/kg) and

**Table 5. Seasonal and site wise variation in Pb, Cd and Ni (mg/Kg) of soil.**

| Parameters | Seasons | Sites | | | | Mean | C.D (P≤0.05) |
|---|---|---|---|---|---|---|---|
| | | Center | Inside | Outside | Control | | |
| **Pb** | Spring | 89.1±3.34[b] | 76.4±1.62[c] | 65.8±2.44[d] | 5.03±0.58[hi] | **59.08[c]** | Sites: 1.799 Seasons: 1.799 Sites × Seasons: 3.557 |
| | Summer | 98.3±4.05[a] | 92.5±2.53[ab] | 87.4±2.25[b] | 10.2±1.22[h] | **72.1[a]** | |
| | Autumn | 96.8±1.69[a] | 88.6±2.09[b] | 81.4±1.56[c] | 7.4±0.90[h] | **68.5[b]** | |
| | Winter | 55.9±1.41[e] | 46.6±2.00[f] | 40.3±1.83[g] | 0.98±0.07[i] | **35.9[d]** | |
| | **Mean** | **85.02[a]** | **76.02[b]** | **68.72[c]** | **5.90[d]** | | |
| **Cd** | Spring | 4.69±1.08[a] | 4.12±1.00[ab] | 3.98±0.85[ab] | 0.12±0.02[cd] | **3.22[b]** | Sites: 1.10 Seasons:1.10 Sites × Seasons: 2.20 |
| | Summer | 6.11±1.28[a] | 5.89±0.76[a] | 5.61±0.66[a] | 0.25±0.08[cd] | **4.46[a]** | |
| | Autumn | 5.11±1.23[a] | 5.01±0.50[a] | 4.89±1.28[a] | 0.19±0.03[cd] | **3.80[ab]** | |
| | Winter | 2.34±0.57[bc] | 2.02±0.43[bcd] | 1.87±0.52[bcd] | 0.06±0.32[d] | **1.57[c]** | |
| | **Mean** | **4.56[a]** | **4.26[a]** | **4.08[a]** | **0.15[b]** | | |
| **Ni** | Spring | 26.20±1.02[de] | 22.50±2.15[ef] | 18.30±1.45[fg] | 1.06±0.24[i] | **17.01[c]** | Sites: 2.047 Seasons: 2.047 Sites × Seasons:4.093 |
| | Summer | 42.10±3.02[a] | 38.70±4.33[ab] | 35.70±2.80[bc] | 2.32±0.46[i] | **29.70[a]** | |
| | Autumn | 32.40±2.11[c] | 29.80±3.51[cd] | 24.90±1.96[de] | 1.85±0.19[i] | **22.23[b]** | |
| | Winter | 15.47±2.17[g] | 12.46±1.64[gh] | 8.65±1.10[h] | 0.65±0.18[i] | **9.30[d]** | |
| | **Mean** | **29.04[a]** | **25.86[b]** | **21.88[c]** | **1.47[d]** | | |
| Permissible value of soil (mg/kg) Pb = 85; Cd = 0.8; Ni = 35 | | | | Permissible value of plant Pb = 2; Cd = 0.02; Ni = 10 | | Permissible value of water Pb = 0.05; Cd = 0.005; Ni = 0.07 | |

Pb: Lead, Cd: Cadmium, Ni: Nickel.

**Table 6. Seasonal and site wise variation in Cr, and As (mg/Kg) of soil.**

| Parameters | Seasons | Sites | | | | Mean | C.D (P≤0.05) |
|---|---|---|---|---|---|---|---|
| | | Center | Inside | Outside | Control | | |
| Cr | Spring | 21.80±1.52[cd] | 19.60±1.62[de] | 16.70±1.49[ef] | 3.41±0.63[i] | 15.37[c] | Sites: 1.43 |
| | Summer | 30.40±3.25[a] | 27.60±1.58[ab] | 25.30±2.01[bc] | 6.34±0.72[hi] | 22.41[a] | Seasons: 1.43 |
| | Autumn | 26.10±1.21[abc] | 24.90±1.95[bc] | 22.80±2.52[cd] | 4.89±1.03[i] | 19.67[b] | Sites × Seasons: 2.86 |
| | Winter | 15.20±1.47[ef] | 12.40±1.54[fg] | 9.64±0.41[gh] | 2.31±0.26[i] | 9.88[d] | |
| | Mean | 23.37[a] | 21.12[a] | 18.61[b] | 4.23[c] | | |
| As | Spring | 13.40±1.12[bcd] | 9.65±0.65[ef] | 8.67±0.42[ef] | 0.45±0.07[h] | 8.04[c] | Sites: 1.31 |
| | Summer | 18.50±1.47[a] | 16.30±1.76[ab] | 14.60±1.45[bc] | 1.12±0.12[h] | 12.63[a] | Seasons: 1.31 |
| | Autumn | 15.30±1.58[bc] | 12.70±1.26[cd] | 10.60±1.15[de] | 0.86±0.05[h] | 9.86[b] | Sites × Seasons:2.62 |
| | Winter | 9.21±0.79[ef] | 7.48±0.68[fg] | 5.62±1.10[g] | 0.23±0.07[h] | 5.63[d] | |
| | Mean | 14.10[a] | 11.53[b] | 9.87[c] | 0.66[d] | | |
| Permissible value of soil (mg/kg) Cr = 100; As = 20 | | | | Permissible value of plant (mg/kg) Cr = 1.30 As = 0.05 | | | Permissible value of Water (mg/kg) Cr: 0.05 As: 0.05 |

Cr: Chromium, As: Arsenic.

min during winter (5.63 mg/kg). Among sites maximum concentration (14.10 mg/kg) was reported at center of dumpsite, followed by inside (11.53 mg/kg), outside (9.87 mg/kg) and minimum was reported at control site (0.66 mg/kg) as depicted in Table 6. The mean values of As between seasons were significantly different. In addition to this the mean values of As between all sites was also reported to be significantly different. Furthermore the correlation between the recorded paramaters is depicted in Table 7.

**Table 7. Correlation between various physico–chemical paramaters of soil.**

| | pH | EC | MC | N | P | K | Ca | Mg | Zn | Fe | Cu | Mn | Pb | Cd | Ni | Cr | As |
|---|---|---|---|---|---|---|---|---|---|---|---|---|---|---|---|---|---|
| pH | 1 | | | | | | | | | | | | | | | | |
| EC | -0.86 | 1 | | | | | | | | | | | | | | | |
| MC | -0.48 | 0.78 | 1 | | | | | | | | | | | | | | |
| N | -0.87 | 0.81 | 0.61 | 1 | | | | | | | | | | | | | |
| P | -0.92 | 0.92 | 0.60 | 0.93 | 1 | | | | | | | | | | | | |
| K | -0.86 | 0.82 | 0.65 | 0.99 | 0.92 | 1 | | | | | | | | | | | |
| Ca | -0.88 | 0.84 | 0.67 | 0.98 | 0.92 | 0.98 | 1 | | | | | | | | | | |
| Mg | -0.87 | 0.84 | 0.67 | 0.98 | 0.92 | 0.98 | 0.99 | 1 | | | | | | | | | |
| Zn | -0.36 | 0.57 | 0.85 | 0.37 | 0.34 | 0.43 | 0.45 | 0.45 | 1 | | | | | | | | |
| Fe | -0.39 | 0.64 | 0.86 | 0.40 | 0.41 | 0.46 | 0.45 | 0.45 | 0.93 | 1 | | | | | | | |
| Cu | -0.33 | 0.46 | 0.80 | 0.39 | 0.28 | 0.46 | 0.47 | 0.46 | 0.95 | 0.88 | 1 | | | | | | |
| Mn | -0.68 | 0.84 | 0.93 | 0.71 | 0.69 | 0.75 | 0.78 | 0.78 | 0.88 | 0.86 | 0.85 | 1 | | | | | |
| Pb | -0.87 | 0.75 | 0.56 | 0.95 | 0.86 | 0.95 | 0.93 | 0.92 | 0.43 | 0.46 | 0.48 | 0.70 | 1 | | | | |
| Cd | -0.84 | 0.67 | 0.43 | 0.93 | 0.83 | 0.92 | 0.89 | 0.87 | 0.31 | 0.33 | 0.36 | 0.57 | 0.98 | 1 | | | |
| Ni | -0.89 | 0.71 | 0.41 | 0.92 | 0.88 | 0.91 | 0.87 | 0.86 | 0.29 | 0.33 | 0.30 | 0.57 | 0.95 | 0.98 | 1 | | |
| Cr | -0.90 | 0.73 | 0.45 | 0.93 | 0.87 | 0.92 | 0.90 | 0.88 | 0.35 | 0.38 | 0.38 | 0.62 | 0.98 | 0.98 | 0.98 | 1 | |
| As | -0.90 | 0.79 | 0.54 | 0.95 | 0.91 | 0.95 | 0.92 | 0.91 | 0.41 | 0.45 | 0.42 | 0.69 | 0.97 | 0.96 | 0.98 | 0.98 | 1 |

## Discussion

Soil pH determines the acidity or alkalinity of soil solutions and refers to the soil's hydrogen ion content. Natural soil typically has a pH between 7 and 8.5, although this range can vary based on biological activity, temperature, and municipal waste disposal practices [28]. Soil pH influences the availability of essential plant nutrients and the concentration of toxic elements, thereby affecting plant growth [29]. As it impacts the availability of all nutrients in the soil it has a direct effect on the survival and development of plants. In this, study pH of dumpsite soils ranged from 6.1 to 6.9 throughout study period. The mean pH values ranged from slightly acidic to neutral, which may be due to the decomposition of organic matter leading to the formation of organic acids. However, the pH increased with increase in distance from dumpsite which may be attributed to release of less organic acids away from dumpsite. Further, the lowest pH value reported during summer season may be attributed to more degradation of organic matter leading to increase in production of organic acids because of increased microbial activity and favorable temperature and moisture conditions during summer season as compared to winter season as depicted in Fig 2.

The pH value of control site (Shalimar) soil was higher than the other three sites during all the seasons, this may be due to presence of less organic matter and hence less microbial activity. The slightly acidic pH of the soils at three other sites suggests that solid waste contributed to soil acidity, leading to a decrease in the availability of essential plant nutrients such as phosphorus and molybdenum, while increasing the availability of toxic elements, particularly aluminum and manganese [30]. Electrical conductivity is an indirect measurement that is correlated with a number of the physical and chemical characteristics of soil. EC is affected by moisture content that soil particles hold. The total quantity of anions and cations determines electrical conductivity. It also depends on the primary salts in the soil solution, which have a negative impact on the qualities of the soil [31]. The optimal EC is crop specific, and depends on environmental conditions [32]. In general, higher EC hinders nutrient uptake by increasing the osmotic pressure of the nutrient solution, and the increased discharged of nutrients into the environment, resulting in environmental pollution. Lower EC may severely affect plant health and yield [33,34]. In the current study, maximum mean value of EC was reported at center of dumping site, which may be due to increased concentration of cations and anions within the dumpsite soil [35] and minimum was reported at control site. Among seasons maximum mean EC value was reported during summer season, which may be due to increase in degradation of salts and release of large number of cations and anions favoured by temperature and minimum was reported during winter due to decrease in degradation of organic matter as shown in Table 1 and Fig 3.

The electrical conductivity of the soil samples collected within the dumpsite was high compared to the soil sample collected from the control site. Moisture content refers to the amount of water soil can hold. Most soil samples from municipal solid waste have moisture content ranging from 15% to 40% [31]. The highest mean moisture content was observed at the center of the dumping site, which may be because dumpsite soil retains more moisture due to the natural ground being covered by municipal solid waste, which has high moisture content and prevents soil moisture from evaporating directly [36]. The mean moisture content of soil decreased with the increase of the distance from the dumpsite. This higher moisture content in the dumpsite soil may be associated with the presence of high organic matter which has capacity to hold more water [37]. This excess amount of water suffocates plant roots by reducing the amount of oxygen available to them. Without enough oxygen, roots are not able to absorb nutrients and water effectively, which in turn inhibits the growth of plants. The moisture content was high during winter season which may be attributed to the high precipitation rate

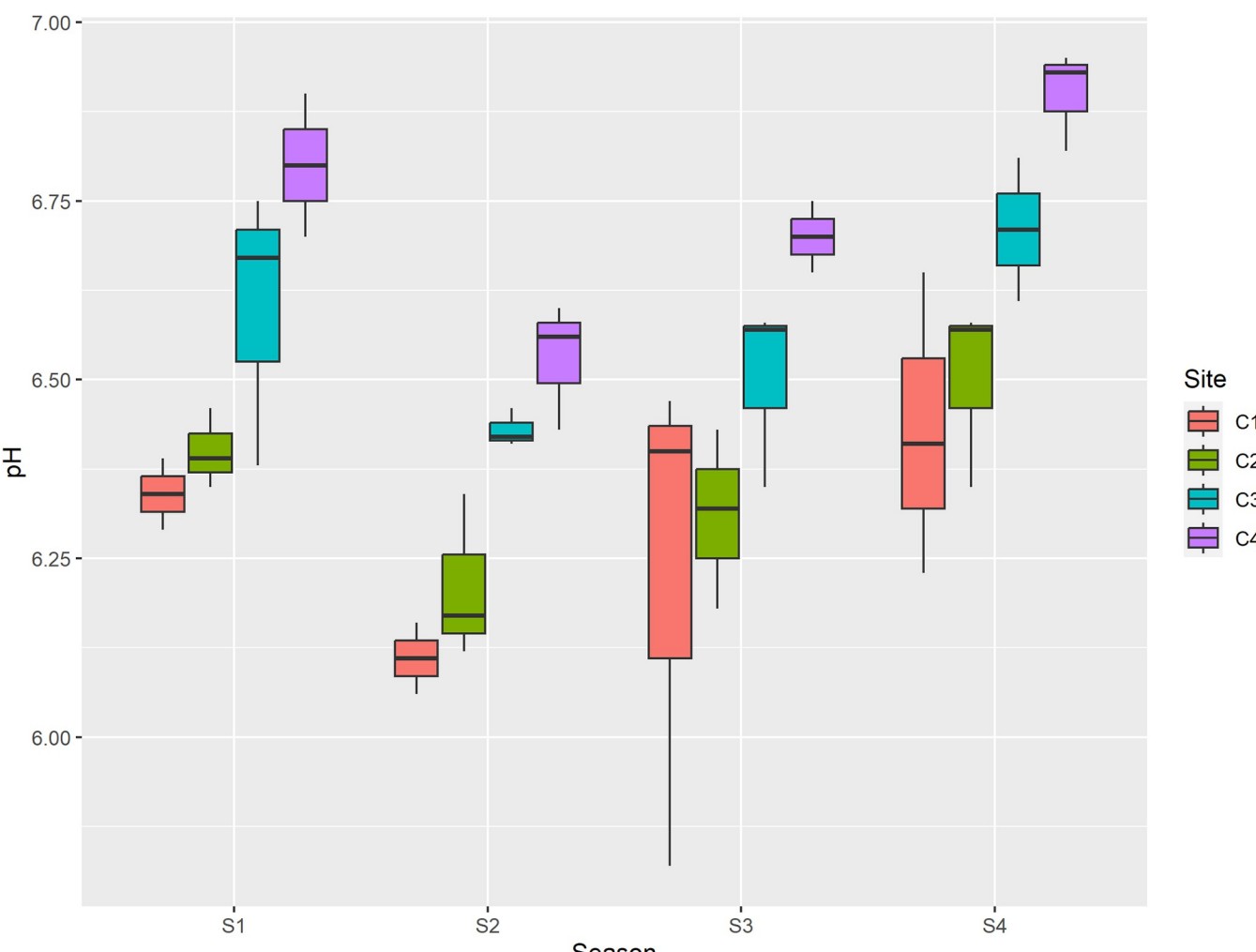

**Fig 2. Graphical representation of pH among different sites and seasons.** S1 is spring, S2 is summer, S3 is autumn and S4 is winter, while as C1 is center, C2 is inside, C3 is outside and C4 is Control.

during winter. Nitrogen (N) is the most important ingredient that plants take up from the soil. In the present study, the concentration of N was maximum at center of dumping site during all the seasons as shown in Table 2. The samples within the dumpsite have high N concentration than the samples away from dumpsite, which may be attributed to abundant amount of the putrescible waste (biowaste, kitchen waste) [38]. The maximum N concentration observed during summer season may be due to increase in reaction rate because of high temperature. Additionally, the high levels of organic matter present in dump soil may be responsible for the high N content [39]. It has been reported that application of N significantly increases tuber yield and other yielding components in Colocasia [40]. Although N is a necessary component for the synthesis of proteins and amino acids, a high concentration of nitrogen in the tissues can be hazardous to the plant's physiological and phonological responses [41]. The movement of leachate from municipal solid waste (MSW) disposal site can increase the phosphorus (P) content in soil at dumpsites. In the current study the highest mean P level was found at the center of the dumping site. e which may be attributed to the large quantities of organic waste, paper and cardboard, textiles, and plastics [42]. Seasonally, the highest mean phosphorus

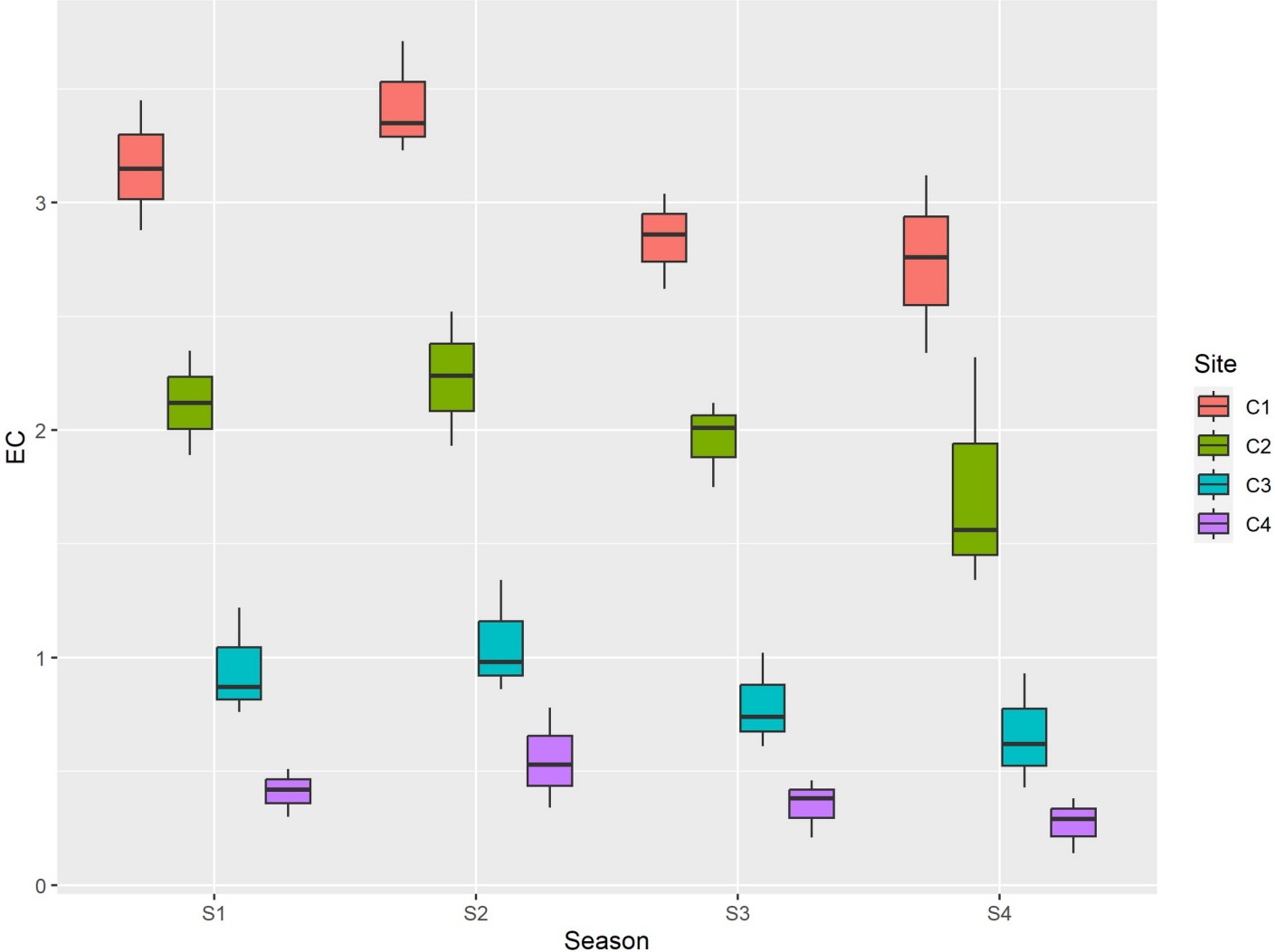

**Fig 3. Graphical representation of EC (dS/m) among different sites and seasons.** S1 is spring, S2 is summer, S3 is autumn and S4 is winter, while as C1 is center, C2 is inside, C3 is outside and C4 is Control.

content was observed during the summer (and the lowest during winter. The concentration of P was higher in the soil samples of the dumping site than the controlled site. This may be explained by the fact that organic matter in waste contains sizable amounts of P, and that orthophosphate is released into the soil solution during mineralization. Additionally, humic substances and organic acids can be absorbed into soil surfaces, leading to depletion in ability of P adsorption by obstructing sites for the formation of complexes with Al, Fe, and Ca. Inorganic and organic products are also produced during the partial decomposition of organic waste [43]. Potassium readily dissolves in water; however, it remains largely intact in undisturbed soils. After being released from decomposing organic matter, potassium quickly and strongly binds to clay particles [31]. In this study the highest concentration of potassium was found at the center of the dumping site, likely due to the decomposition of a substantial amount of kitchen waste. Conversely, the lowest concentration was observed at the control site, where there was less organic waste as given in Table 2. Among seasons, maximum K content reported during summer may be due to high decomposition rate of kitchen waste and less

leaching and minimum content of it reported during winter may be due to high leaching losses by precipitation. Similar results were obtained by Deshmukh and Aher [28]. Despite the fact that K controls stomatal opening and shutting, cell enlargement, and other critical physiological activities. Numerous research have examined importance of potassium on plant growth, however high K concentrations in soil solutions restrict magnesium uptake, leading to magnesium deficiency in plants [44]. Calcium (Ca) is crucial element in figuring out the soil's physical and chemical properties, especially its structure and pH. Clay platelets in soil colloids (caused by calcium ions) bind or aggregate to produce crumbs or peds. Frequent correlation has been reported between pH and exchangeable calcium levels in soil. For example in the pH range of 7.0 to 8.5, Ca is accessible, while as low Exchangeable calcium levels were reported in acidic pH [31]. In the current study maximum Ca content reported during summer may be because of degradation of calcium rich waste such as bones and eggshells present in the kitchen waste and minimum was reported during winter, which may be because of low rate of degradation of waste. Among sites, maximum Ca content reported at center of dumping site may be attributed to the clayey nature of soils of dumpsite as shown in Table 3. Magnesium (Mg) is found in both relatively soluble forms and ionic form, $Mg^{2+}$, which is attached to the soil colloidal complex. It has been reported that magnesium levels in acidic soils, especially sandy soils, are frequently low, while as neutral soils usually contain more exchangeable magnesium [31]. In this investigation among sites maximum Mg concentration reported at center of dumping site may be because of its less mobility due to presence of high organic waste and clayey nature of soil [45] and minimum at control site may be because of the presence of less organic matter as given in Table 3. Among seasons maximum Mg concentration reported during summer season may be beca use of less leaching and high rate of decomposition of organic waste and minimum during winter may be attributed to high leaching loss due to the high positive water balance [45]. An excessive amount of magnesium in the soil inhibits plant growth because it reduces the ability of roots to reduce TTC (2, 3, 5-Triphenyl Tetrazolium chloride) through the inhibition of the dehydrogenase enzyme linked to the respiratory systems [46].

Micronutrients are essential for sustaining the fertility of the soil and crop productivity. Compared to macronutrients, micronutrients are required in smaller amounts. Even if the supply of macronutrients is balanced and high yielding varieties are cultivated, lack of micronutrients will prevent maximum yield from being achieved [47]. In this study, the presence of Zn element in both naturally occurring materials, such as yard wastes and food wastes, as well as man-made materials like pigments, inks, metals, and plastics may account for the highest mean Zn value at the centre of the dumping site, while the lowest mean Zn value was reported at the control site, as shown in Table 4. Furthermore micronutrients are more tightly bonded to the soil at high pH than at low pH, the highest mean Zn concentration during the winter season may be attributed to low pH, whereas the lowest mean Zn concentration during the summer season may be attributed to high pH [48]. Although Zn serves as a nutrient for plants, but is hazardous to them in larger amounts. Zn toxicity in plants results in growth inhibition, a reduction in biomass production, competition for nutrients, inactivation of enzymes, removal of vital components from functional locations, chlorosis, and, in certain situations, blockage of Fe translocation [49].

Ferric oxide ($Fe_2O_3$), also known as hematite, is the most prevalent form of Iron (Fe) in soils. It is exceedingly insoluble and gives soil its red colour. Commonly, the oxide form is hydrated [50]. In this study the maximum mean Fe concentration recorded at the centre of the dumping site may be attributed to the presence of organic matter in soils, as organic matter rich soils contain Fe in the reduced state ($Fe^{2+}$) or is adsorbed on the surface of soil particle [50]. The maximum iron concentration reported during winter season depicted in Table 4 may be attributed to low pH, because with increase in pH, ferrous ion ($Fe^{2+}$) gets converted to

ferric ion ($Fe^{3+}$). Low solubility in solution makes the ferric ion ($Fe^{3+}$) compounds less bio-available, and a minimum concentration was recorded during the summer [51]. The root cells and plasma membrane are damaged by the extra iron ions, which results in oxidative stress and increased formation of reactive oxygen species (ROS). Root cells may die as a result of ROS harming cellular components such lipids, proteins, and DNA [52]. Furthermore it, can also affect the uptake of other essential elements, such as phosphorus and zinc, by competing with the transporters that facilitate their uptake by the roots. This interference can lead to nutrient imbalances, reduced growth, and overall plant health [53].

Compared to other trace metals, copper (Cu) has extremely little mobility because it is firmly adsorbed onto soil particles. Due to this reduced mobility, Cu has a tendency to concentrate on soil particles. Leaching of Cu occurs when the amount of Cu in the soil exceeds the capacity of the soil type to hold the copper ions. The pH, cation exchange capacity, organic matter content, and presence of iron, manganese, and aluminium oxides all affect how much Cu is present in soils. The amount of water in the soil affects the ability of the soil to store copper through biotic and abiotic oxidation-reduction reactions [31]. Among sites mean maximum concentration reported at center of dumping site may be due to the higher stability constants of Cu complexes with organic matter and minimum was reported at control site as shown in Table 4. Among seasons, maximum Cu concentration reported during winter season may be because of low pH, as at low pH Cu compounds are more soluble and minimum concentration of Cu reported during summer may be because of high pH, as high pH results in precipitation of copper (II) hydroxide (Cu ($OH$)$_2$) and Copper(I) hydroxide (CuOH) [54]. All phases of plant growth are affected by high Cu concentration toxicity, which has considerable detrimental impacts on morphology, physiological function, and molecular level [55]. Additionally, too much copper in the soil inhibits the activity of -amylase and invertase isoenzymes, which limits the ability of many plant species to break down reserve food supplies like starch and sucrose [56]. Maximum concentration of Mn reported at center of dumping site may be due to decomposition of plants and animals waste as well as animal excrement [57]. The lowest concentration of Mn was reported at control site as given in Table 4. The highest Mn concentration observed during the winter season, is likely due to a decrease in soil pH. At low pH, trace elements are typically more soluble because of high desorption and low adsorption [31]. Conversely, the lowest Mn concentration was reported during the summer, which may be attributed to increased adsorption [58]. Manganese is essential for terrestrial plants, but excessive accumulation in leaves can cause toxicity and reduce crop yield.

Heavy metals are significant pollutants that enter in to environment through various channels, including landfill leachate, solid waste, and wastewater. According to Radfard et al. [59] and Kusin et al. [60], heavy metals in leachate contaminate soil and subsurface water supplies. As these contaminants move up the food chain, they impact human health and cause various ailments. Leachate is a critical source of pollution, releasing chemicals into soil and water sources, and its toxicity to human cells highlights its environmental impact [61]. In the current study among seasons max mean concentration of Pb was reported during summer season and min during winter. Among sites maximum concentration was reported at center of dumpsite and minimum was reported at control site. The values of Pb are higher than permissible limit (85mg/kg) at center during spring, summer and autumn, inside during summer and autumn and outside during summer season as depicted in Table 5 and Fig 4.

Lead-acid batteries used in automobiles, rechargeable nickel-cadmium batteries, consumer electronics, glass, ceramics, and plastic products including PVC resins are sources of Pb in landfill leachate. Other sources of lead include soldered cans, pigments, brass and bronze products, rubber goods, spent motor oil, and wine bottle wrappers made of lead foil. Among seasons max concentration of Cd was reported during summer season and min during winter.

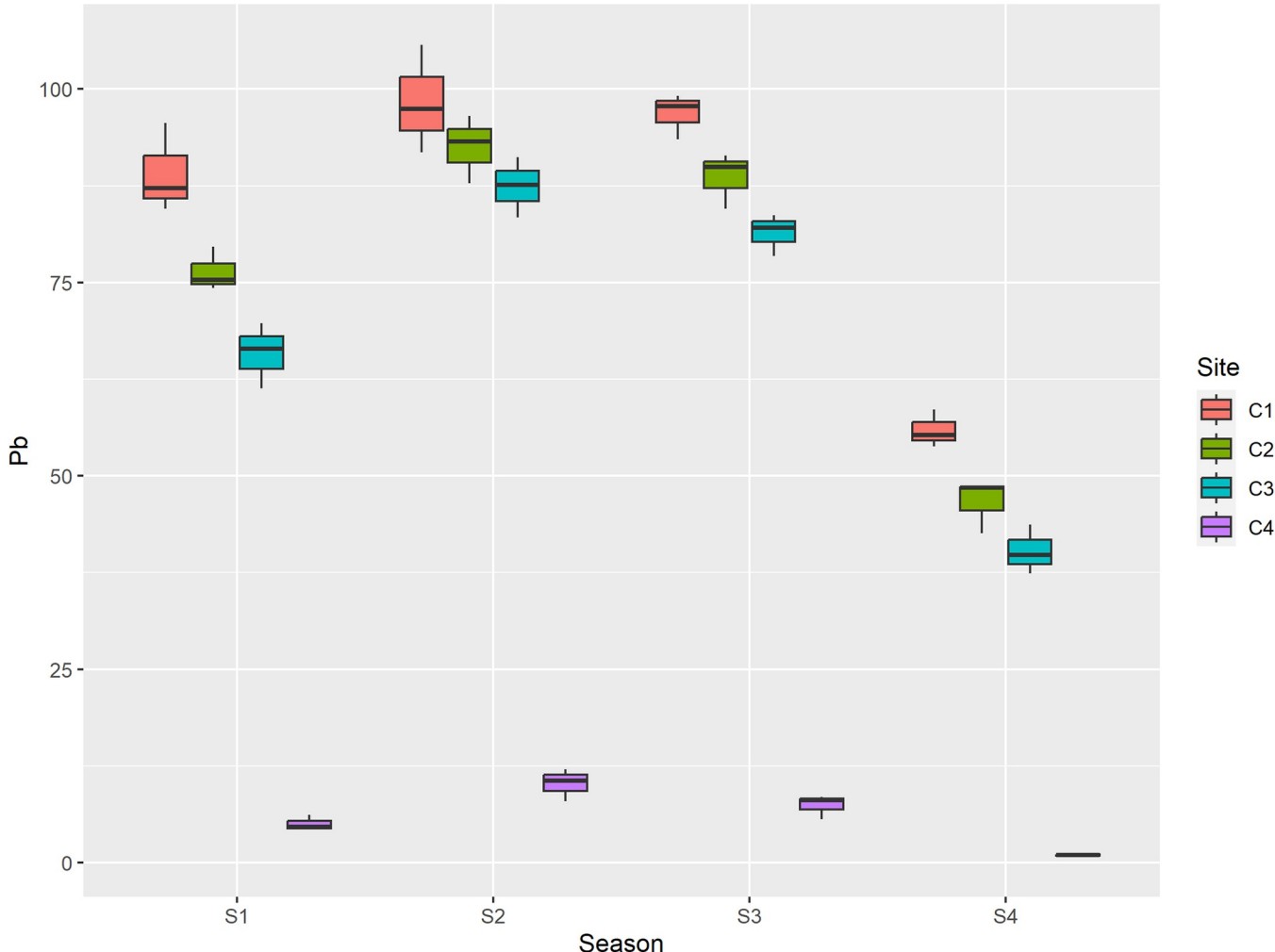

**Fig 4. Graphical representation of Pb (mg/kg) among different sites and seasons.** S1 is spring, S2 is summer, S3 is autumn and S4 is winter, while as C1 is center, C2 is inside, C3 is outside and C4 is Control.

Among sites maximum concentration was reported at center of dumpsite, and minimum was reported at control site. Cadmium (Cd) levels at all sites exceeded the permissible limit of 0.8 mg/kg across all seasons as depicted in Table 5 and Fig 5.

The sources of cadmium in waste include nickel-cadmium batteries, pigments, stabilizers in plastics (primarily PVC), consumer electronics (notably the steel chassis of vintage TVs and radios that were cadmium-plated to prevent corrosion), antique appliances with cadmium-plated parts, textile dyes and paints, glass and ceramics, and pigments used in non-newspaper printing inks. Cadmium is one of the most ecotoxic metals, causing detrimental effects on soil health, plant metabolism, and the health of humans and animals [62]. Even at very low concentrations, chronic exposure to cadmium can lead to anemia, anosmia, cardiovascular diseases, and renal problems [63]. Ni was reported maximum during summer season and min during winter. Among sites maximum concentration was reported at center of dumpsite and minimum was reported at control site as depicted in Table 5 and Fig 6.

The values of Ni were higher than the permissible limits (35 mg/kg) during summer season at all sites. The values were lower than the permissible limits during other seasons. The sources

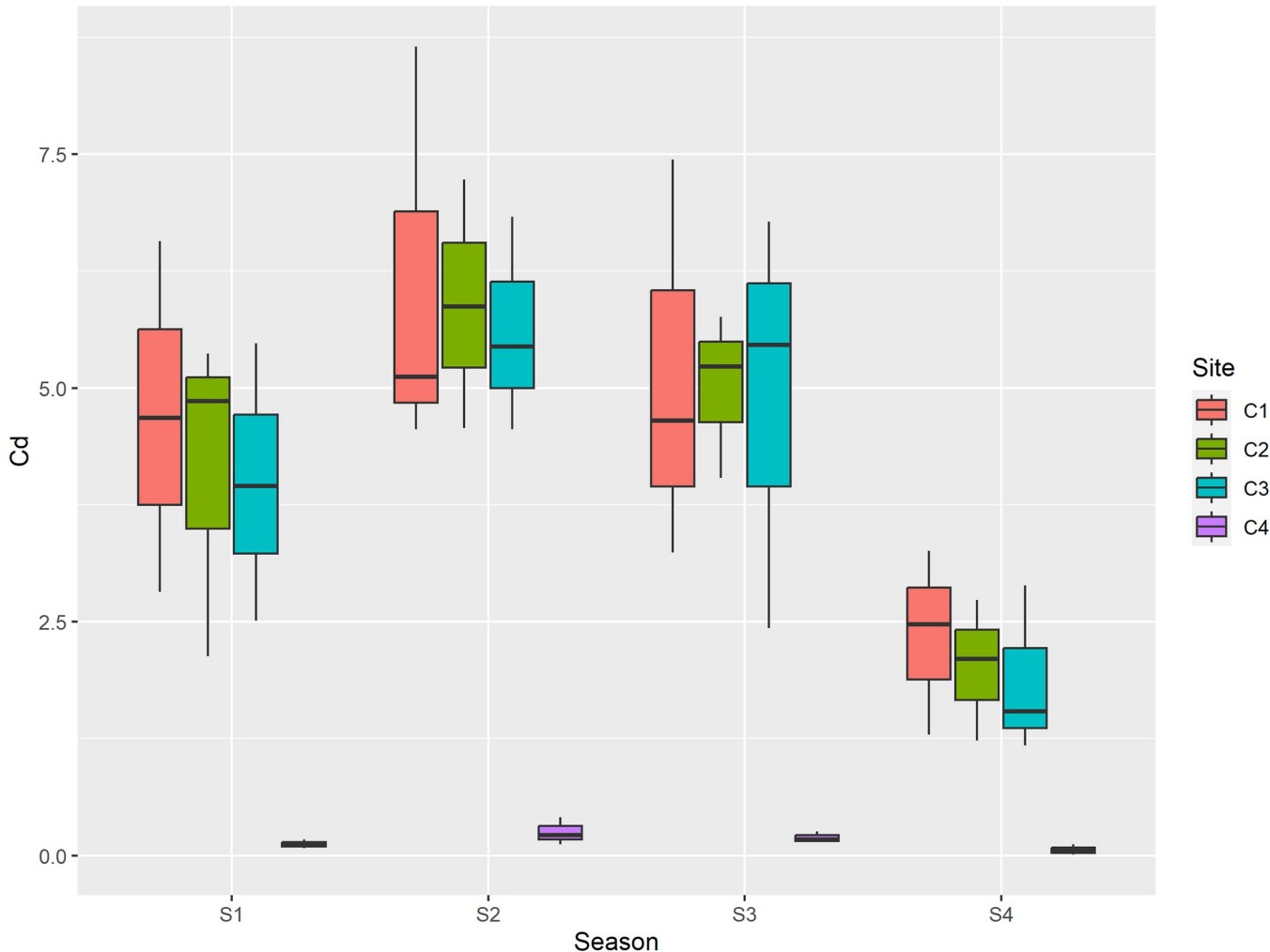

**Fig 5. Graphical representation of Cd (mg/kg) among different sites and seasons.** S1 is spring, S2 is summer, S3 is autumn and S4 is winter, while as C1 is center, C2 is inside, C3 is outside and C4 is Control.

of Ni are lead cadmium batteries, electroplating materials, clinical wastes [64]. Because nickel is non-biodegradable and has the potential to accumulate in the food web, it is responsible for a number of health issues in humans, including allergic skin, headache, vertigo, nausea, vomiting, insomnia, and irritability [64,65]. Among seasons max concentration of Cr was reported during summer season and min during winter. Among sites maximum concentration was reported at center of dumpsite, and minimum was reported at control site as depicted in Table 6 and Fig 7.

The sources of Cr are electroplating and Cr containing wastes. The concentration of Cr was less than permissible value at all sites. The sorption properties of the soil, such as its clay, iron oxide, and organic matter contents, influence the mobility of Cr. Surface runoff can carry chromium to surface waters in both its soluble and precipitated forms. Chromium compounds, both soluble and unadsorbable, can seep into groundwater from the soil. As soil pH rises, Cr (VI) has a greater propensity to leach. However, the majority of Cr discharged into natural rivers is particle associated and eventually deposits into the sediment [66]. Human allergic dermatitis and chromium are related [67]. The value of Arsenic (As) ranged from 0.23

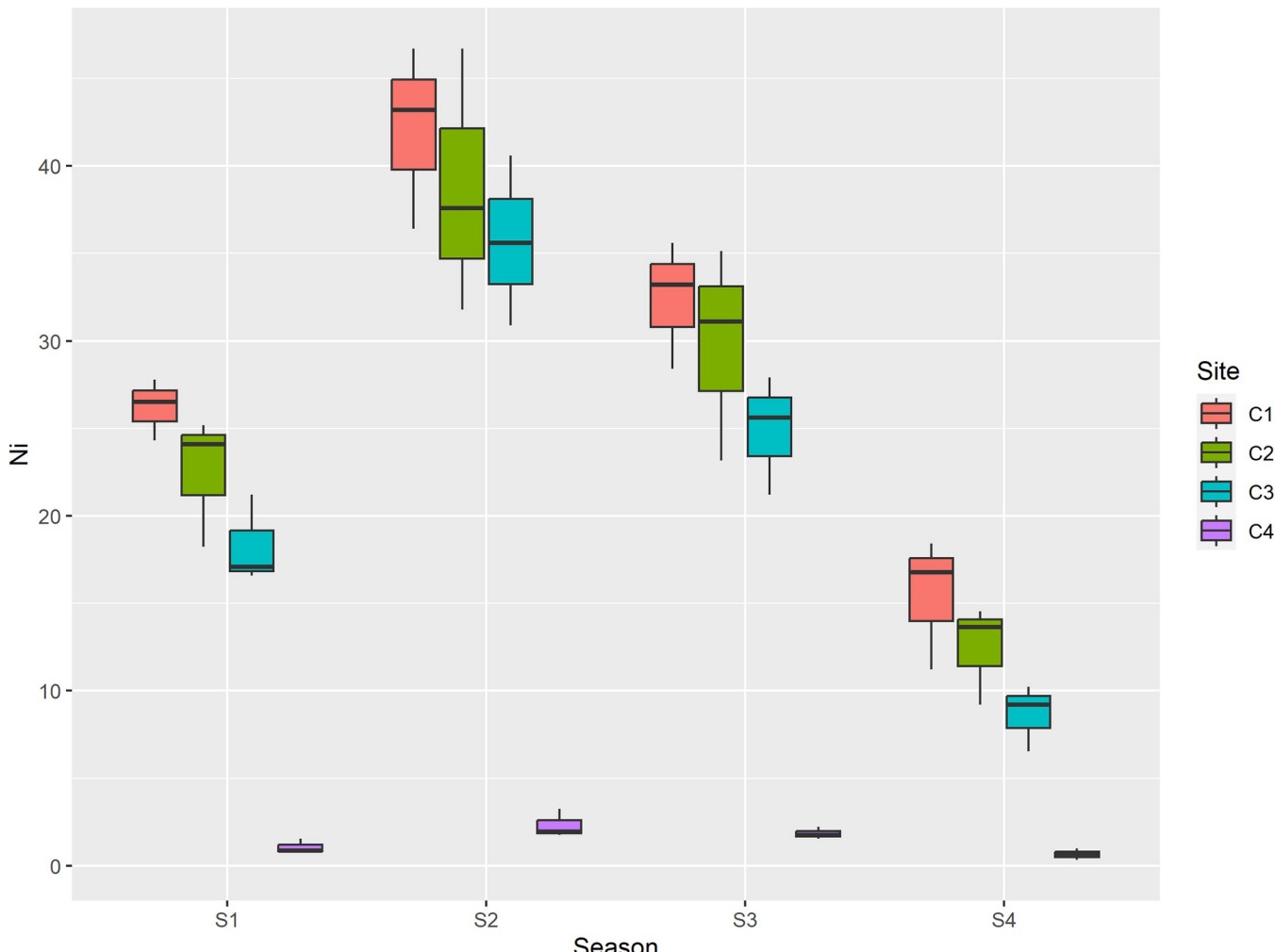

**Fig 6. Graphical representation of Ni (mg/kg) among different sites and seasons.** S1 is spring, S2 is summer, S3 is autumn and S4 is winter, while as C1 is center, C2 is inside, C3 is outside and C4 is Control.

mg/kg to 18.50 mg/kg. Among seasons max concentration was reported during summer season and min during winter. Among sites maximum concentration was reported at center of dumpsite, and minimum was reported at control site as depicted in Table 6 and Fig 8.

The permissible limit of As in soil is 20 mg/kg. The concentration of heavy metals, specifically Pb, Cd, Ni, Cr, and As, is higher during the summer than other seasons. The leaching of heavy metals is also higher during the summer because temperature influences it; as the temperature rises, the leaching of heavy metals also increases. However, the minimum concentration of heavy metals during winter season might be attributed to runoff effect during winter season which facilitates the leaching of heavy metals from soil and contributes to the dilution of soil solution during winter season.

Heavy metal toxicity in plants severely hampers nutrient and water uptake, while intensifying oxidative damage, ultimately stunting plant growth. Cadmium (Cd) has been shown to interact with plants at the physio-biochemical level, leading to reduced growth. Cd toxicity disrupts nutrient and water absorption, heightens oxidative stress, and impairs plant metabolism, and negatively impacts plant morphology and physiology [68]. Similarly, chromium (Cr)

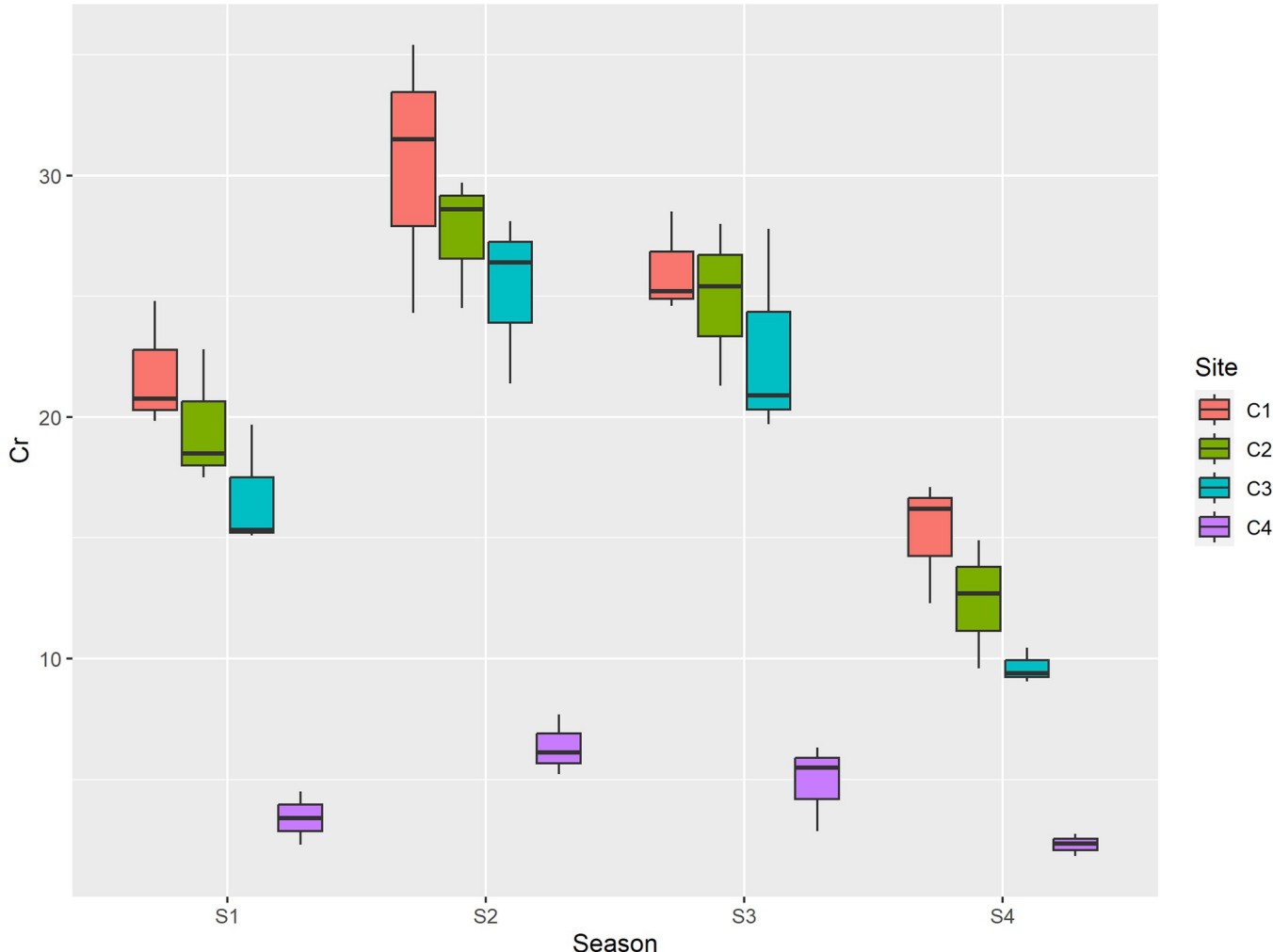

**Fig 7. Graphical representation of Cr (mg/kg) among different sites and seasons.** S1 is spring, S2 is summer, S3 is autumn and S4 is winter, while as C1 is center, C2 is inside, C3 is outside and C4 is Control.

toxicity inhibits seed germination, slows plant growth, impairs enzymatic activities, and disrupts the photosynthetic machinery, leading to oxidative imbalances [69,70]. Lead (Pb) toxicity has been found to inhibit ATP production, elevate reactive oxygen species (ROS) levels, and cause DNA damage. It significantly affects germination, root elongation, plant growth, chlorophyll synthesis, and disturbs transcriptome sequencing [71]. Nickel (Ni) toxicity reduces seed germination, root and shoot development, biomass accumulation, and overall yield. It also causes chlorosis, necrosis, and disrupts physiological processes like photosynthesis and transpiration, while inducing oxidative stress. The threat posed by Ni increases with its concentration in the environment, particularly in soils [72]. Arsenic (As) exposure adversely affects plants at both biochemical and molecular levels, inhibiting key physiological processes such as overall growth, photosynthetic efficiency, and biomass production. Arsenic induces oxidative stress by increasing ROS production or reducing their elimination, leading to damage of lipids, proteins, and nucleic acids. It interferes with metabolic pathways, either by acting as a competitive inhibitor of phosphate (Pi) or by disrupting enzyme activities. Additionally, As inhibits seed germination, root and shoot development, and other critical early-stage processes during seedling growth [73].

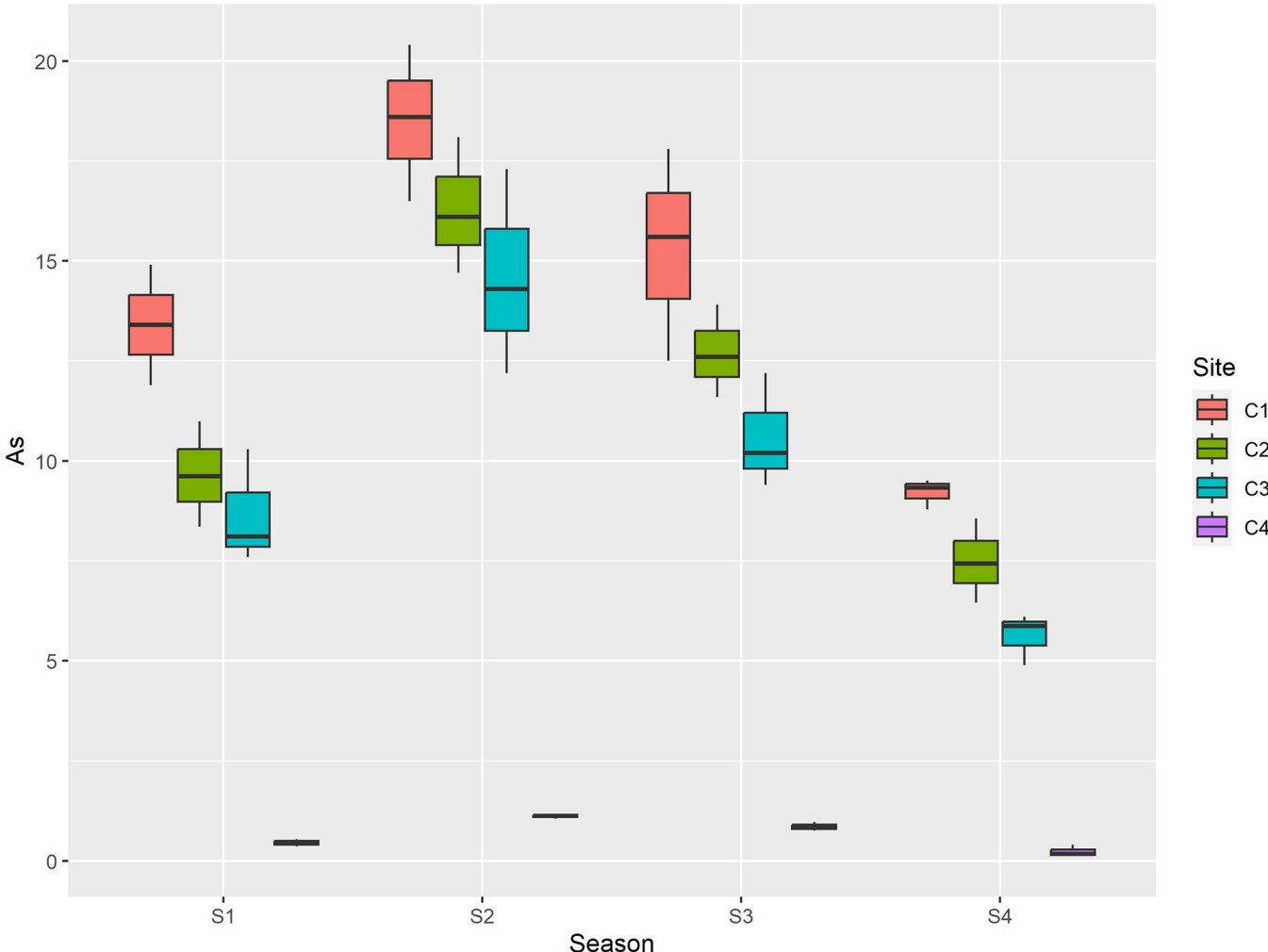

**Fig 8. Graphical representation of As (mg/kg) among different sites and seasons.** S1 is spring, S2 is summer, S3 is autumn and S4 is winter, while as C1 is center, C2 is inside, C3 is outside and C4 is Control.

## Available management strategy for remediation of soil health

Soil is very important asset for plant and human health, so the need of hour is to maintain the soil quality and prevent its degradation. The waste generated from various sources like houses, restaurants, agricultural fields, orchards is mostly organic (65–75%), which is diverted to landfill in huge amounts on daily basis can be segregated at source and then converted into compost. The compost generated can then be used to ameliorate the nutrient status of soil quality as well as to enhance the plant growth and development. Furthermore the compost generated will replace the chemical fertilizers used. The plan of utilization of microbial consortium technology for degradation of waste is depicted in flow chart given in Fig 9.

## Conclusion

The soil plays a crucial role in controlling the flow of pollutants into the food chain. Population growth is causing waste generation to grow rapidly as well. Landfills have been a popular choice for residue dumping for the past 20 years. Our investigation has shown that, regardless of the time of year or the distance from the waste site, landfills seriously degrade the soil quality of nearby

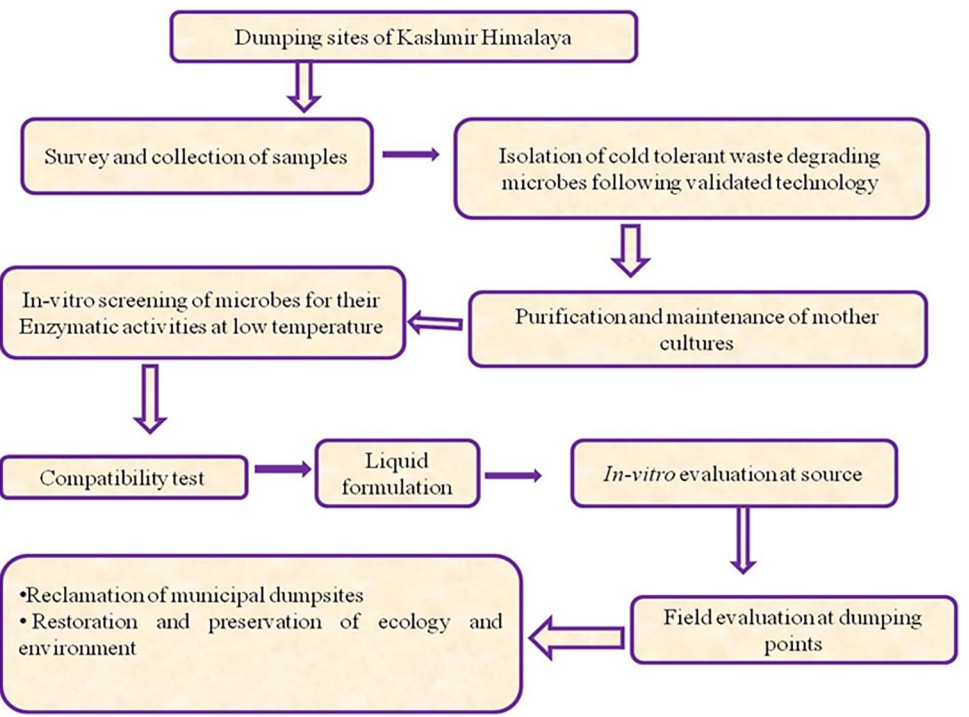

**Fig 9. Flow chat depicting management strategy for waste.**

communities. pH was more acidic during the summer season and at center of the dumping site due to an increase in the production of organic acids, while as it was less acidic during the winter season and away from the dump site. The acidic pH results in a decrease in the availability of essential elements viz, N.P and K, however, it results in an increase in the availability of toxic heavy metals viz, Pb, Cd, Ni, Cr, As. The heavy metals were reported to be maximum during the summer season at all sites and minimum during the winter season. Among sites the heavy metals were maximum and decreased with increase in distance from the landfill. Our findings led to the conclusion that there is a need for global attention on improving or restoring soil health. Assessment of soil health indicators is expected to enhance our understanding of the factors underlying processes that contribute to sustainable agriculture. The leachate must be treated before disposal and that landfills must have adequate lining. Before disposing of rubbish, segregate non-biodegradable items like polythene and plastic bags since they take roughly 500 years to degrade. Biodegradable garbage must be handled using ecologically friendly techniques like anaerobic digestion and composting. The garbage that needs to be landfilled should only be non-biodegradable, inert, or unsuitable for recycling. MSW must not be combined with biomedical waste. Media should be used to inform people about trash management. The organic waste ought to be isolated at the point of generation and turned into compost using microbial processes.

## Supporting information

**S1 Data.**
(XLSX)

## Acknowledgments

The authors acknowledge the support of SKUAST Kashmir and Division of Environmental sciences in carrying out the research.

## Author Contributions

**Conceptualization:** Haleema Bano.

**Formal analysis:** Shayesta Islam.

**Funding acquisition:** Fahad Alotaibi.

**Software:** Shayesta Islam.

**Supervision:** Haleema Bano, Asif Aziz Malik.

**Writing – original draft:** Shayesta Islam.

**Writing – review & editing:** Haleema Bano, Asif Aziz Malik, Fahad Alotaibi.

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
