## [Decision Letter · Decision Letter 0]

21 Jun 2024

PONE-D-24-19939Landfill leachate: an invisible threat to soil quality of temperate HimalayasPLOS ONE

Dear Dr. Malik,

Thank you for submitting your manuscript to PLOS ONE. After careful consideration, we feel that it has merit but does not fully meet PLOS ONE’s publication criteria as it currently stands. Therefore, we invite you to submit a revised version of the manuscript that addresses the points raised during the review process.

We look forward to receiving your revised manuscript.

Kind regards,

Susmita Lahiri (Ganguly)

Academic Editor

PLOS ONE

6. We note that Figure 1 in your submission contain [map/satellite] images which may be copyrighted. All PLOS content is published under the Creative Commons Attribution License (CC BY 4.0), which means that the manuscript, images, and Supporting Information files will be freely available online, and any third party is permitted to access, download, copy, distribute, and use these materials in any way, even commercially, with proper attribution. For these reasons, we cannot publish previously copyrighted maps or satellite images created using proprietary data, such as Google software (Google Maps, Street View, and Earth). For more information, see our copyright guidelines: http://journals.plos.org/plosone/s/licenses-and-copyright.

Reviewers' comments:

Reviewer's Responses to Questions

**Comments to the Author**

1. Is the manuscript technically sound, and do the data support the conclusions?

Reviewer #1: Yes

Reviewer #2: Yes

2. Has the statistical analysis been performed appropriately and rigorously? 

Reviewer #1: Yes

Reviewer #2: Yes

3. Have the authors made all data underlying the findings in their manuscript fully available?

Reviewer #1: Yes

Reviewer #2: Yes

4. Is the manuscript presented in an intelligible fashion and written in standard English?

Reviewer #1: Yes

Reviewer #2: Yes

5. Review Comments to the Author

Reviewer #1: Comments

I have read the exhaustive article on “Landfill leachate: an invisible threat to soil quality of temperate Himalayas”. In this article, the authors thoroughly document previous literature. However, it is necessary to cite the following research articles in the introduction and methodology sections before publishing.

1. DOI: 10.1007/s11104-023-06016-4

2. DOI: 10.3390/microorganisms10102078

3. DOI: 10.4236/ojss.2018.82007

Reviewer #2: PAPER PRESENTS NOVEL WORK HOWEVER IT REQUIRES MINOR EDITING IN INTRODUCTION AND DISCUSSION

Landfills are the most affordable and popular method for managing waste in many

parts of the world, However, in most developing nations, including India, the infiltration

of hazardous materials from improperly managed dumping sites continues to be a

significant environmental problem. Around the world, leachate is a significant point

source of contamination in numerous environmental media, including soil,

groundwater, and surface water.

6. PLOS authors have the option to publish the peer review history of their article (what does this mean?). If published, this will include your full peer review and any attached files.

Reviewer #1: No

Reviewer #2: **Yes: **Owais Ali Wani

---

## [Author Response · Author response to Decision Letter 0]

21 Jul 2024

Rebuttal

Response: Formatted my manuscript as per PLOS ONE templates attached (Title, author affiliations and main body).

Response: I have included the full name of authority that permitted for visit to the study site in my Methods section.

Authors do not need to submit their entire data set if only a portion of the data was used in the reported study. If your submission does not contain these data, please either upload them as Supporting Information files or deposit them to a stable, public repository and provide us with the relevant URLs, DOIs, or accession numbers. For a list of recommended repositories, please see https://journals.plos.org/plosone/s/recommended-repositories. 

Response: The raw data of results of this study has been attached as separate file “Supplementary data”. There are no ethical or legal restrictions in sharing the data. 

Response: The ORCID iD of the corresponding author has been validated in Editorial Manager.

Response: The ethical statement was earlier in acknowledgement section. Now I have moved to Methods section.

6. We note that Figure 1 in your submission contain [map/satellite] images which may be copyrighted. All PLOS content is published under the Creative Commons Attribution License (CC BY 4.0), which means that the manuscript, images, and Supporting Information files will be freely available online, and any third party is permitted to access, download, copy, distribute, and use these materials in any way, even commercially, with proper attribution. For these reasons, we cannot publish previously copyrighted maps or satellite images created using proprietary data, such as Google software (Google Maps, Street View, and Earth). For more information, see our copyright guidelines: http://journals.plos.org/ plosone/s/licenses-and-copyright. 

We recommend that you contact the original copyright holder with the Content Permission Form (http://journals.plos.org/ plosone/s/file?id=7c09/content-permission-form.pdf) and the following text: “I request permission for the open-access journal PLOS ONE to publish XXX under the Creative Commons Attribution License (CCAL) CC BY 4.0 (http://creativecommons.org/licenses/by/4.0/). Please be aware that this license allows unrestricted use and distribution, even commercially, by third parties. Please reply and provide explicit written permission to publish XXX under a CC BY license and complete the attached form.” 

NASA Earth Observatory (public domain): http://earthobservatory.nasa.gov/ Landsat: http://landsat.visibleearth.nasa.gov/

USGS EROS (Earth Resources Observatory and Science (EROS) Center) (public domain): http://eros.usgs.gov/# Natural Earth (public domain): http://www.naturalearthdata.com/

Response: I have revised map as per suggestions and have used resources from the sources which don’t have copyright issues.

Response: I have reviewed the reference list as per the journal requirements. Few reference including 22, 24, 25, 26, 27, 33, 34, 35, 40, 41, 51. The reference Hochmuth, 2017 has been replaced by Lucena and Apaolaza, 2017. 

5. Review Comments to the Author 

Please use the space provided to explain your answers to the questions above. You may also include additional comments for the author, including concerns about dual publication, research ethics, or publication ethics. (Please upload your review as an attachment if it exceeds 20,000 characters).

Reviewer #1: Comments 

I have read the exhaustive article on “Landfill leachate: an invisible threat to soil quality of temperate Himalayas”. In this article, the authors thoroughly document previous literature. However, it is necessary to cite the following research articles in the introduction and methodology sections before publishing. 

1. DOI: 10.1007/s11104-023-06016-4 

Response: Cited from line number 52-54, reference no 6.

2. DOI: 10.3390/microorganisms10102078 

Response: Cited from line number 63-64, reference no 10.

3. DOI: 10.4236/ojss.2018.82007

Response: Cited from line number 250-251, reference no 41.

Reviewer #2: 

PAPER PRESENTS NOVEL WORK HOWEVER IT REQUIRES MINOR EDITING IN INTRODUCTION AND DISCUSSION Landfills are the most affordable and popular method for managing waste in many parts of the world, However, in most developing nations, including India, the infiltration of hazardous materials from improperly managed dumping sites continues to be a significant environmental problem. Around the world, leachate is a significant point source of contamination in numerous environmental media, including soil, groundwater, and surface water.

Response: Done as per suggestions

Response: Uploaded all the figure files in PACE.

---

## [Decision Letter · Decision Letter 1]

23 Aug 2024

PONE-D-24-19939R1Landfill leachate: an invisible threat to soil quality of temperate HimalayasPLOS ONE

Dear Dr. Malik,

Thank you for submitting your manuscript to PLOS ONE. After careful consideration, we feel that it has merit but does not fully meet PLOS ONE’s publication criteria as it currently stands. Therefore, we invite you to submit a revised version of the manuscript that addresses the points raised during the review process.

We look forward to receiving your revised manuscript.

Kind regards,

Susmita Lahiri (Ganguly)

Academic Editor

PLOS ONE

Journal Requirements:

Reviewers' comments:

Reviewer's Responses to Questions

**Comments to the Author**

1. If the authors have adequately addressed your comments raised in a previous round of review and you feel that this manuscript is now acceptable for publication, you may indicate that here to bypass the “Comments to the Author” section, enter your conflict of interest statement in the “Confidential to Editor” section, and submit your "Accept" recommendation.

Reviewer #2: All comments have been addressed

Reviewer #3: All comments have been addressed

Reviewer #4: All comments have been addressed

Reviewer #5: All comments have been addressed

Reviewer #6: (No Response)

2. Is the manuscript technically sound, and do the data support the conclusions?

Reviewer #2: Yes

Reviewer #3: Yes

Reviewer #4: Yes

Reviewer #5: Yes

Reviewer #6: Yes

3. Has the statistical analysis been performed appropriately and rigorously? 

Reviewer #2: Yes

Reviewer #3: Yes

Reviewer #4: Yes

Reviewer #5: Yes

Reviewer #6: Yes

4. Have the authors made all data underlying the findings in their manuscript fully available?

Reviewer #2: Yes

Reviewer #3: Yes

Reviewer #4: Yes

Reviewer #5: Yes

Reviewer #6: Yes

5. Is the manuscript presented in an intelligible fashion and written in standard English?

Reviewer #2: Yes

Reviewer #3: Yes

Reviewer #4: Yes

Reviewer #5: Yes

Reviewer #6: Yes

6. Review Comments to the Author

Reviewer #2: (No Response)

Reviewer #3: Dear Authors,

The subject of the study is interesting and topical, with scientific and practical importance.

The authors took into account the recommendations of the reviewers and made the necessary corrections.

The article has been considerably improved.

The following aspects are brought to the attention of the authors.

1.Article structure

Page 13, row 101

“Study area:”

Page 14

“Methodology:”

It is recommended to check and revise the structure of chapters (chapter titles), according to the one recommended in the Submission Guidelines, PLOS ONE.

e.g.

“Manuscript Organization”

“Materials and Methods”

2.Writing ionic forms

Page 27, row 348

“Mg2+” instead of “Mg2+”

"2+" as index

Page 30, row 402

“Fe2O3” instead of “Fe2O3”

Page 31, row 404

“Fe2+” instead of “Fe++”

"2+" as index

row 409

“Fe3+” instead of “Fe3+”

"3+" as index

It is recommended to use the Word Equation Editor to write the ionic forms correctly

3.Presentation of the Figures

Figure 14 is presented on page 41

Until figure 14, no other figure is presented in the content of the article.

References are made to the other figures, but they are presented at the end, starting with page 53.

Please check and revise, if necessary, according to the Submission Guidelines, PLOS ONE

4. References

Please check the References chapter, and revise, if necessary, according to the Submission Guidelines, PLOS ONE

Reviewer #4: PLOS ONE- Landfill leachate: an invisible threat to soil quality of temperate Himalayas

Manuscript Number: PONE-D-24-19939R1

Overall comments

The manuscript is informative. I am very positive about the manuscript, but it still requires additional work and minor editing before being considered for publication.

Please ensure that your manuscript meets PLOS ONE's style requirements. The paper needs a total revision; especially the results section and the discussion section should be separate.

Please review your reference list to ensure that it is complete and correct. Please follow the references style of plos journal formet. Provide DOI no. of the references and try to add recent references. For example ‘Amoakwah E, Arthur E, Frimpong KA, Islam KR. Biochar Amendment Influences Tropical Soil Carbon and Nitrogen Lability. Journal of Soil Science and Plant Nutrition. 2021; 21: 3567–3579. http://dx.doi. org/10.1007/s42729-021-00628-4’’

Specific comments

1. Please check no. of words in Abstract, not exceed 300 words according to journal style. Please ensure that your manuscript meets PLOS ONE's style requirements

2. Please insert the title of the figure or table just below the text. Please ensure that your manuscript meets PLOS ONE's style requirements

3. It would be better to separate the results section and the discussion section. Please check the PLOS ONE's style.

4. Rewrite the discussion section.

5. Please follow the reference style of PLOS ONE's style. Provide DOI no. of the references and try to add recent references. Please review your reference list to ensure that it is complete and correct. For example ‘’Amoakwah E, Arthur E, Frimpong KA, Islam KR. Biochar Amendment Influences Tropical Soil Carbon and Nitrogen Lability. Journal of Soil Science and Plant Nutrition. 2021; 21: 3567–3579. http://dx.doi. org/10.1007/s42729-021-00628-4’’

6. Please check the front style of headline, sub-headline and text front. Please check the PLOS ONE's style.

7. If possible try to reduce no of figures. No. of fig 14 is too high.

8. Please check line no. 58. Remove double full stop.

9. Please check line no. 84. Close bracket

10. Please check line no. 90. Check the space

11. Please check line no. 108. Remove bracket

12. Please check line no. 121. Edit space

13. Please check line no. 121, 150, 121. Edit space

14. Please check line no. 312, 314, 352, and 354 and so on. Why are the numbers bold?

15. Please check Table 3, column 8. Some words are overlapped

16. Please check line no. 474, 476,506. Please check the line spacing

17. Please provide declaration, consent for publication and conflict of interest from the manuscript and add in a separate file to PLOS ONE's online system. Only Acknowledgement and Author contributions would be attached with the article.

18. Please ensure that your manuscript meets PLOS ONE's style requirements.

Reviewer #5: A brief study on "Landfill leachate: an invisible threat to soil quality of temperate

Himalayas" has been conducted by the authors. The major concern is the content of toxic elements like Pb, Cd, Ni, As and Cr in the leachate. This will help the environmentalist and planner of the region for sustainable development goals for the locality.

Please mention the safe limit of all the parameters studied in this work for soil just like Pb, Cd, Ni, As and Cr.

Reviewer #6: 1. Abstract: Grammatical correction is required in lines- 15, 19, 29, 30, 33-36, 41. Punctuation correction is required in lines-30, 34.

2. Introduction: Too much narration on soil health in the introduction section (lines 54-64) while less description of the keywords -landfill and leachate. An individual paragraph on Achan landfill and leachate could be helpful for better understanding and reading. Coordination of the paragraphs in the introduction section is suggested. Grammatical and punctuation correction is required in all through the introduction section (lines- 45-47, 52,54,58, 60-68, 81, 82).

3. Study area: No indication of the area in the sampling site. Precise information about the sampling site is needed. Grammatical and punctuation correction is required in the study area section- 108,112,113

4. Methodology: No indication of year of sampling. Why sampling was done in those locations? Are those sites the leachate outflow sites? Authors analyzed only soils. But analysis of both soil and leachate would give a clear understanding about the impact of landfill leachate on soil quality. Grammatical and punctuation correction is required in the methodology section (lines- 120-122, 140)

5. Result and discussion: I suggest to establish a relationship among pH, EC and element concentration.

7. PLOS authors have the option to publish the peer review history of their article (what does this mean?). If published, this will include your full peer review and any attached files.

Reviewer #2: No

Reviewer #3: No

Reviewer #4: **Yes: **Dr. Rakhi Rani Sarker

Senior Scientific Officer

Soil Science Division

Bangladesh Institute of Nuclear Agriculture,

Mymensingh-2202, Bangladesh

E-mail: rrssarker@gmail.com

Reviewer #5: No

Reviewer #6: **Yes: **Prof. Dr. Md. Rafiqul Islam

---

## [Author Response · Author response to Decision Letter 1]

18 Sep 2024

Review Comments to the Author Please use the space provided to explain your answers to the questions above. You may also include additional comments for the author, including concerns about dual publication, research ethics, or publication ethics. (Please upload your review as an attachment if it exceeds 20,000 characters) 

Reviewer #2: (No Response) 

Reviewer #3: 

Dear Authors, 

The subject of the study is interesting and topical, with scientific and practical importance. The authors took into account the recommendations of the reviewers and made the necessary corrections. The article has been considerably improved. 

The following aspects are brought to the attention of the authors. 

1.Article structure 

Page 13, row 101 

“Study area:” 

Page 14 

“Methodology:” 

It is recommended to check and revise the structure of chapters (chapter titles), according to the one recommended in the Submission Guidelines, PLOS ONE. 

e.g. 

“Manuscript Organization” 

“Materials and Methods” 

Response: All the chapter titles and subtitles have been revised according to the guidelines of PLOS ONE

2.Writing ionic forms 

Page 27, row 348 

“Mg2+” instead of “Mg2+” 

"2+" as index 

Page 30, row 402 

“Fe2O3” instead of “Fe2O3” 

Page 31, row 404 

“Fe2+” instead of “Fe++” 

"2+" as index 

row 409 

“Fe3+” instead of “Fe3+” 

"3+" as index 

It is recommended to use the Word Equation Editor to write the ionic forms correctly 

Response: All the corrections suggested have been made and word equation editor has been used to write ionic forms. 

3. Presentation of the Figures 

Figure 14 is presented on page 41 

Until figure 14, no other figure is presented in the content of the article. 

References are made to the other figures, but they are presented at the end, starting with page 53. 

Please check and revise, if necessary, according to the Submission Guidelines, PLOS ONE 

Response: According to submission guidelines of PLOS ONE, figures are not to be included in the main manuscript file, figures are to be prepared as individual file. The Fig 14 has now also been removed from main manuscript. The captions of all figures have been inserted in the text of manuscript immediately after the paragraph in which it is cited. 

4. References Please check the References chapter, and revise, if necessary, according to the Submission Guidelines, PLOS ONE 

Response: References have been revised according to submission guidelines

Reviewer #4: PLOS ONE- Landfill leachate: an invisible threat to soil quality of temperate Himalayas 

Manuscript Number: PONE-D-24-19939R1 

Overall comments �

The manuscript is informative. I am very positive about the manuscript, but it still requires additional work and minor editing before being considered for publication. 

Please ensure that your manuscript meets PLOS ONE's style requirements. The paper needs a total revision; especially the results section and the discussion section should be separate.

Response: Separated result and discussion section 

 Please review your reference list to ensure that it is complete and correct. Please follow the references style of plos journal formet. Provide DOI no. of the references and try to add recent references. For example ‘Amoakwah E, Arthur E, Frimpong KA, Islam KR. Biochar Amendment Influences Tropical Soil Carbon and Nitrogen Lability. Journal of Soil Science and Plant Nutrition. 2021; 21: 3567– 3579. http://dx.doi. org/10.1007/s42729-021-00628-4’’ 

Response: Revised reference section according to journal guidelines

Specific comments 

1. Please check no. of words in Abstract, not exceed 300 words according to journal style. Please ensure that your manuscript meets PLOS ONE's style requirements 

Response: Abstract has been reduced to 300 words only

2. Please insert the title of the figure or table just below the text. Please ensure that your manuscript meets PLOS ONE's style requirements 

Response: Title of figures and tables have been inserted in the text as per journal guidelines. 

3. It would be better to separate the results section and the discussion section. Please check the PLOS ONE's style. 

Response: Separated results and discussion section. 

4. Rewrite the discussion section. 

Response: Modified the discussion section

5. Please follow the reference style of PLOS ONE's style. Provide DOI no. of the references and try to add recent references. Please review your reference list to ensure that it is complete and correct. For example ‘’Amoakwah E, Arthur E, Frimpong KA, Islam KR. Biochar Amendment Influences Tropical Soil Carbon and Nitrogen Lability. Journal of Soil Science and Plant Nutrition. 2021; 21: 3567– 3579. http://dx.doi. org/10.1007/s42729-021-00628-4’’

Response: Revised reference section as per journal guidelines 

6. Please check the front style of headline, sub-headline and text front. Please check the PLOS ONE's style. 

Response: Followed PLOS one font style for each section.

7. If possible try to reduce no of figures. No. of fig 14 is too high.

Response: Reduced the number of figures to 9 

8. Please check line no. 58. Remove double full stop. 

Response: Done

9. Please check line no. 84. Close bracket 

Response: Done

10. Please check line no. 90. Check the space 

Response: Done

11. Please check line no. 108. Remove bracket 

Response: Done

12. Please check line no. 121. Edit space 

Response: Done

13. Please check line no. 121, 150, 121. Edit space 

Response: Done

14. Please check line no. 312, 314, 352, and 354 and so on. Why are the numbers bold? 

Response: Done

15. Please check Table 3, column 8. Some words are overlapped 

Response: The line numbers have been overlapped with the table values, corrected

16. Please check line no. 474, 476,506. Please check the line spacing 

Response: Done

17. Please provide declaration, consent for publication and conflict of interest from the manuscript and add in a separate file to PLOS ONE's online system. Only Acknowledgement and Author contributions would be attached with the article. 

Response: Done as per suggestion

18. Please ensure that your manuscript meets PLOS ONE's style requirements. 

Response: Modified as per journal guidelines

Reviewer #5: A brief study on "Landfill leachate: an invisible threat to soil quality of temperate Himalayas" has been conducted by the authors. The major concern is the content of toxic elements like Pb, Cd, Ni, As and Cr in the leachate. This will help the environmentalist and planner of the region for sustainable development goals for the locality. Please mention the safe limit of all the parameters studied in this work for soil just like Pb, Cd, Ni, As and Cr. 

Response: Safe limits for all paramaters have been incorporated

Reviewer #6: 

1. Abstract: Grammatical correction is required in lines- 15, 19, 29, 30, 33-36, 41. Punctuation correction is required in lines-30, 34. 

Response: Corrected

2. Introduction: Too much narration on soil health in the introduction section (lines 54-64) while less description of the keywords - landfill and leachate. An individual paragraph on Achan landfill and leachate could be helpful for better understanding and reading. Coordination of the paragraphs in the introduction section is suggested. Grammatical and punctuation correction is required in all through the introduction section (lines- 45-47, 52,54,58, 60-68, 81, 82). 

Response: Line numbers from 54-64 have been removed from the manuscript. Separate paragraph on Achan landfill and leachate have been added. Grammatical and punctuation errors have been corrected.

3. Study area: No indication of the area in the sampling site. Precise information about the sampling site is needed. Grammatical and punctuation correction is required in the study area section- 108,112,113 

Response: Area under landfill have been added and corrections are made.

4. Methodology: No indication of year of sampling. Why sampling was done in those locations? Are those sites the leachate outflow sites? Authors analyzed only soils. But analysis of both soil and leachate would give a clear understanding about the impact of landfill leachate on soil quality. Grammatical and punctuation correction is required in the methodology section (lines- 120-122, 140) 

Response: Mentioned year of sampling and reasons for soil sampling, Grammatical and punctuation correction done

5. Result and discussion: I suggest to establish a relationship among pH, EC and element concentration. 

Response: Incorporated correlation table in results section (Table 7)

Response: Done

---

## [Decision Letter · Decision Letter 2]

29 Oct 2024

PONE-D-24-19939R2Landfill leachate: an invisible threat to soil quality of temperate HimalayasPLOS ONE

Dear Dr. Malik,

Thank you for submitting your manuscript to PLOS ONE. After careful consideration, we feel that it has merit but does not fully meet PLOS ONE’s publication criteria as it currently stands. Therefore, we invite you to submit a revised version of the manuscript that addresses the points raised during the review process.

We look forward to receiving your revised manuscript.

Kind regards,

Susmita Lahiri (Ganguly)

Academic Editor

PLOS ONE

Journal Requirements:

Reviewers' comments:

Reviewer's Responses to Questions

**Comments to the Author**

1. If the authors have adequately addressed your comments raised in a previous round of review and you feel that this manuscript is now acceptable for publication, you may indicate that here to bypass the “Comments to the Author” section, enter your conflict of interest statement in the “Confidential to Editor” section, and submit your "Accept" recommendation.

Reviewer #3: All comments have been addressed

Reviewer #6: All comments have been addressed

2. Is the manuscript technically sound, and do the data support the conclusions?

Reviewer #3: Yes

Reviewer #6: Yes

3. Has the statistical analysis been performed appropriately and rigorously? 

Reviewer #3: Yes

Reviewer #6: Yes

4. Have the authors made all data underlying the findings in their manuscript fully available?

Reviewer #3: Yes

Reviewer #6: Yes

5. Is the manuscript presented in an intelligible fashion and written in standard English?

Reviewer #3: Yes

Reviewer #6: Yes

6. Review Comments to the Author

Reviewer #3: Dear Authors,

The subject of the study is interesting and topical, with scientific and practical importance.

The authors took into account the recommendations of the reviewers and made the necessary corrections.

The article has been considerably improved.

The following minor aspects are brought to the attention of the authors.

1.

Space between the bibliographic source number and the previous word

e.g.

"Gomez [16]."

2.

Central positioning of all values in the table columns.

e.g.

Table 1

3.

Space between the value and the unit of measure

e.g.

"0.98 mg/kg"

Some suggestions were made in the article.

Reviewer #6: The authors have revised the manuscript as per my comments and suggestions. Now the manuscript has become sound scientifically.

7. PLOS authors have the option to publish the peer review history of their article (what does this mean?). If published, this will include your full peer review and any attached files.

Reviewer #3: No

Reviewer #6: **Yes: **Dr. Md. Rafiqul Islam

---

## [Author Response · Author response to Decision Letter 2]

30 Oct 2024

6. Review Comments to the Author

Reviewer #3: Dear Authors,

The subject of the study is interesting and topical, with scientific and practical importance.

The authors took into account the recommendations of the reviewers and made the necessary corrections.

The article has been considerably improved.

The following minor aspects are brought to the attention of the authors.

1.

Space between the bibliographic source number and the previous word

e.g.

"Gomez [16]."

Reply: Corrected

2.

Central positioning of all values in the table columns.

e.g.

Table 1

Reply: Done

3.

Space between the value and the unit of measure

e.g.

"0.98 mg/kg"

Reply: Done

Some suggestions were made in the article.

Reply: Incorporated all suggestions 

Reviewer #6: The authors have revised the manuscript as per my comments and suggestions. Now the manuscript has become sound scientifically.

---

## [Editor Report · Decision Letter 3]

5 Nov 2024

Landfill leachate: an invisible threat to soil quality of temperate Himalayas

PONE-D-24-19939R3

Dear Dr. Malik,

We’re pleased to inform you that your manuscript has been judged scientifically suitable for publication and will be formally accepted for publication once it meets all outstanding technical requirements.

Kind regards,

Susmita Lahiri (Ganguly)

Academic Editor

PLOS ONE
---

## [Editor Report · Acceptance letter]

8 Nov 2024

PONE-D-24-19939R3 

PLOS ONE

Dear Dr. Malik, 

I'm pleased to inform you that your manuscript has been deemed suitable for publication in PLOS ONE. Congratulations! Your manuscript is now being handed over to our production team.

Kind regards, 

on behalf of

Dr. Susmita Lahiri (Ganguly) 

Academic Editor

PLOS ONE